# Neural Collapse is Globally Optimal in Deep Regularized ResNets and Transformers

**Peter Súkeník**
Institute of Science and Technology (ISTA)
Austria
peter.sukenik@ista.ac.at

**Christoph H. Lampert**[*]
Institute of Science and Technology (ISTA)
Austria
chl@ista.ac.at

**Marco Mondelli**[*]
Institute of Science and Technology (ISTA)
Austria
marco.mondelli@ista.ac.at

## Abstract

The empirical emergence of neural collapse—a surprising symmetry in the feature representations of the training data in the penultimate layer of deep neural networks—has spurred a line of theoretical research aimed at its understanding. However, existing work focuses on data-agnostic models or, when data structure is taken into account, it remains limited to multi-layer perceptrons. Our paper fills both these gaps by analyzing modern architectures in a data-aware regime: we prove that global optima of deep regularized transformers and residual networks (ResNets) with LayerNorm trained with cross entropy or mean squared error loss are approximately collapsed, and the approximation gets tighter as the depth grows. More generally, we formally reduce any end-to-end large-depth ResNet or transformer training into an equivalent unconstrained features model, thus justifying its wide use in the literature even beyond data-agnostic settings. Our theoretical results are supported by experiments on computer vision and language datasets showing that, as the depth grows, neural collapse indeed becomes more prominent.

## 1 Introduction

In 2020, Papyan et al. [43] discovered a surprising geometric structure in learned representations of deep neural networks (DNNs) at convergence. This structure—dubbed "neural collapse" (NC)—was present in various architectures trained on many computer vision datasets, and it concerns the representations of the training samples in the last layer of the network: the feature vectors of the samples from the same class converge to the respective class-mean (NC1); the class-means form a simplex equiangular tight frame (ETF), maximizing the pairwise angles (NC2); finally, the class-means align with the rows of the weight matrix of the last layer (NC3). Similar structures were also subsequently discovered for class-imbalanced classification [51], regression [1] and language modeling [62], demonstrating that neural collapse is ubiquitous when training deep models.

The NC phenomenon raised significant interest in the machine learning community from both theoreticians and practitioners, due to its high relevance in both areas. Theoreticians use it to improve generalization understanding [64, 12, 59] both in-distribution and in transfer learning, OOD detection [15], imbalanced learning understanding [67], theory of feature learning [28, 40], robustness [47], as well as representation learning itself [41, 2, 5, 11, 56]. In practice, neural collapse has implications on

---

[*]Equal contribution

39th Conference on Neural Information Processing Systems (NeurIPS 2025).

transfer learning [32, 55, 6], OOD detection [65, 63, 37], compression [26], performance improvement [55, 9] and other aspects [70, 33, 35].

In accordance with the high relevance of NC, a plethora of works aimed at understanding its origins in DNN training. To this goal, Mixon et al. [42] introduced a simplified mathematical framework, called the "unconstrained features model" (UFM). In the UFM, one assumes the features of the last layer to be free variables and optimizes over them together with the weight matrix of the last layer. Using the UFM, the optimality of the NC has been proven, as well as its emergence during gradient descent training in various settings, see Section 2 for details. However, the UFM has since been criticized [23] for being too simplistic: the gradient dynamics in the UFM are inconsistent with those in the entire DNN trained end-to-end, and the global optima might be misaligned due to the difference between plain Frobenius norm regularization and the representation cost of the features, influenced by the training data. This led to attempts of proving NC in end-to-end training. However, the results so far cover shallow (up to three layer) networks [27, 19, 61] or come with strong assumptions [64, 44, 46, 58, 3, 23] (see Section 2). Moreover, with the exception of [58], all works only focus on multi-layer perceptrons (MLPs). However, NC is equally present and important in modern architectures, such as ResNets ([18]) or transformers ([54]) [58, 62]. The addition of modern DNN components, such as residual connections, layer normalization or attention layers, makes the loss landscape significantly different and thus it is unlikely that the theoretical tools developed so far will be easily adjustable to these newer architectures.

In this work, we fill both mentioned gaps at once. First, we analyze ResNets with LayerNorm and transformers. We are the first to theoretically analyze NC in transformer architectures, while also significantly extending the knowledge on ResNets. Second, our results prove end-to-end approximate optimality of NC in training with weight regularization. This has only ever been done for MLPs with deep linear heads in [23]. To be more precise, our contributions are summarized below.

- For ResNets and transformers with one linear layer per MLP block and constant regularization strength, we prove that NC is the asymptotically optimal solution as the number of blocks goes to infinity. Moreover, all global optima in deep-enough networks must be approximately collapsed and the distance from perfect collapse is non-asymptotically upper-bounded in terms of the depth. These results hold for both cross entropy (CE) and mean squared error (MSE) loss, under minimal assumptions on the data.

- We prove the same set of results for ResNets and transformers with *two* linear layers per MLP block and vanishing regularization strength.

- We support these findings by experiments on computer vision datasets with both ResNets and vision transformers, which show that the amount of collapse increases with the depth of the architecture, as predicted by our theory.

- More generally, we provide a formal connection between deep ResNets/transformers and unconstrained features models: we prove that, as these architectures become deeper, their global optima converge to those of an equivalent UFM. This result holds for a wide class of continuous losses.

Let us highlight the conceptual relevance of the last contribution, which reduces trained DNNs to an equivalent UFM. As a consequence, if one can solve the underlying UFM and identify its global optima (which we do for CE and MSE loss), these optima will be provably approached by globally optimal ResNets and transformers trained end-to-end, as long as they are deep enough. This provides a theoretical justification for the use of UFM in the analysis of these architectures and, in fact, it is the first such justification with a theoretical backing appearing in the literature.

## 2   Related work

**Unconstrained features model (UFM).** First introduced in [42, 10], the UFM has been widely analyzed in the literature. The optimality of NC in the UFM has been proved for CE loss [60, 39, 30], MSE loss [68] and other losses [69]. A line of work [10, 51, 20, 7] has focused on the class-imbalanced setting, formulating a generalized NC geometry and proving its optimality. The loss landscape of the UFM was shown to be benign in [71, 24, 68], and the emergence of NC in the UFM through gradient descent training was proved in [42, 16, 24, 57]. Several extensions of the UFM to non-standard settings have been considered, including GNNs [28], large number of classes [25], unconstrained features regressed to the input data [53] and regression [1]. Recently, the UFM

has been used to describe a form of NC in language modeling, where each context (sample) can be followed by multiple continuations, making the labels effectively stochastic [50, 66]. NC has been considered also for more layers following empirical observations [17, 45, 22, 11] and accordingly, UFM was generalized to multiple linear layers in [8, 14, 36], two non-linear layers in [52] and multiple non-linear layers in [48, 49, 13].

**Beyond UFM.** Going towards the analysis of neural networks trained end-to-end, conditions on data that make NC feasible in the shallow case are identified in [19]. Two-layer networks are considered in [27], which uses NTK theory and other kernel methods to conclude that NC in this regime is rather restricted. To the contrary, in the mean-field regime, positive results about NC1 are given in [61] for certain three-layer networks. In the deep case, convergence to NC is studied in [46, 44, 64, 23]. However, a block-structured empirical NTK is assumed in [46], and symmetric quasi-interpolation is required in [44, 64]. The former does not justify this assumption, while the latter requires an unusual weight regularization and interpolators with a given norm. Wide networks are considered in [23], which proves the emergence of NC1 requiring at least the last two layers to be linear (and even deeper linear heads for NC2 and NC3).

Closer to the scope of the current work, NC is studied in ResNets in [58]. Two main claims are proved: the monotonicity of NC1-NC2 metrics across layers of ResNets, and a negative result about collapse in a variant of UFM similar to the one considered in [53]. However, the monotonicity is proved under the strong assumption that the data evolves across layers on a geodesic, which is not possible in general since one can construct configurations where samples from different classes would collide. Moreover, the UFM taken into account is based on a heuristic derivation (a link between representation cost and transport cost of the features) that does not hold exactly in practice.

## 3 Preliminaries

**Notation.** We study two different data formats and architectures. For ResNets, the input data and one-hot labels are $X_0 \in \mathbb{R}^{d_0 \times N}$ and $Y \in \mathbb{R}^{K \times N}$, where $d_0$ is the input dimension, $N$ the number of samples and $K$ the number of classes. For transformers, the input data and one-hot labels are $X_0 \in \mathbb{R}^{N \times V \times C}$ and $Y \in \mathbb{R}^{N \times K \times C}$, where $C$ is the context length (number of tokens in the prompt) and $V$ the vocabulary size (number of distinct tokens). We take $C = 1$ when the third dimension of $X_0, Y$ is not used. If we index a matrix with three abstract indices, the last one is implicitly equal to 1. We assume a class-balanced setting, i.e., $NC = Kn$, where $n$ is the number of samples per class. Unless stated otherwise, we use $x_{ki}$ to indicate the $i$-th sample of the $k$-th class. For transformers, a sample corresponds to the position of each individual token and, thus, $x_{ki}$ corresponds to a token position labeled as class $k$, with samples ordered arbitrarily. For additional notation regarding vision transformers, see Appendix B.

**ResNets and transformers.** Let $\sigma$ denote the ReLU function. Denote by $\mathrm{LN}(\cdot)$ the output of a normalization layer that first subtracts the mean of each column of the input from itself and then divides each column by its standard deviation (if the input is a vector, it returns the normalized vector; if the input is a matrix or tensor, it returns the matrix or tensor with centered and normalized columns of the inner-most dimension matrices). Define also $\mathrm{id}(\cdot)$ as the identity mapping.

**Definition 3.1.** *An $L$-block ResNet with LayerNorm and one linear layer per block (later referred to as L-RN1) is defined as*

$$f_\theta = \mathrm{lin}_L \circ \mathrm{LN} \circ (\mathrm{id} + \sigma \circ \mathrm{lin}_{L-1}) \circ \mathrm{LN} \circ (\mathrm{id} + \sigma \circ \mathrm{lin}_{L-2}) \circ \cdots \circ \mathrm{LN} \circ (\mathrm{id} + \sigma \circ \mathrm{lin}_1) \circ \mathrm{LN} \circ \mathrm{lin}_0, \quad (1)$$

*where $\mathrm{lin}_l(x) = W_l x + b_l$ for all $l \in \{0, \ldots, L-1\}$ and for $l = L$ we remove the bias term. $\theta$ is the collection of all learnable parameters. We denote as $X_1 = \mathrm{LN}(W_0 X_0 + b_0)$, $X_{l+1} = \mathrm{LN}(X_l + \sigma(W_l X_l + b_l))$ ($l \in \{1, \ldots, L-1\}$), $f_\theta(X_0) = X_{L+1} := W_L X_L$ the intermediate representations of the training data stored in a matrix form. We assume that all intermediate representations $X_l$ ($l \in \{1, \ldots, L\}$) are of dimension d. Analogously, L-RN2 denotes a ResNet with two linear layers per block defined as*

$$f_\theta = \mathrm{lin}_L \circ \mathrm{LN} \circ (\mathrm{id} + \mathrm{lin}_{L-1,2} \circ \sigma \circ \mathrm{lin}_{L-1,1}) \circ \cdots \circ \mathrm{LN} \circ (\mathrm{id} + \mathrm{lin}_{1,2} \circ \sigma \circ \mathrm{lin}_{1,1}) \circ \mathrm{LN} \circ \mathrm{lin}_0, \quad (2)$$

*with $X_1 = \mathrm{LN}(W_0 X_0 + b_0)$, $X_{l+1} = \mathrm{LN}(X_l + W_{l,2}\sigma(W_{l,1} X_l + b_{l,1}) + b_{l,2})$ ($l \in \{1, \ldots, L-1\}$) and $f_\theta(X_0) = X_{L+1} := W_L X_L$.*

**Definition 3.2.** *An L-block transformer with one or two linear layers in the attention sub-block and one or two layers in the MLP sub-block (later referred to as L-T11, L-T12, L-T21, L-T22 based on the number of linear layers in attention and MLP sub-blocks, respectively) is defined as*

$$f_\theta(Z) = \lin_{L+1} \circ \mathrm{LN}_{L+1} \circ \mathrm{B}_L \circ \cdots \circ \mathrm{B}_1 \circ \mathrm{Embed}(Z). \tag{3}$$

*Here,* $\lin_{L+1}(Z) = W_{L+1}Z$ *is the last layer;* $\mathrm{Embed}(Z) = W_e Z + W_p$ *is the embedding layer with* $W_e$ *being the token embedding and* $W_p$ *(having the same shape as* $W_e Z$*) the positional embedding; and the* $l$*-th block is given by*

$$\mathrm{B}_l = \mathrm{MLP}_l \circ \mathrm{LN}_{l,2} \circ \mathrm{ATTN}_l \circ \mathrm{LN}_{l,1} . \tag{4}$$

*Such block consists of the normalization layers* $\mathrm{LN}_{l,1}, \mathrm{LN}_{l,2}$*, the MLP*

$$\mathrm{MLP}_l(Z) = Z + \sigma(W_l Z + b_l), \ \text{or} \ \mathrm{MLP}_l(Z) = Z + W_{l,2}\sigma(W_{l,1}Z + b_{l,1}) + b_{l,2}, \tag{5}$$

*respectively for the architecture L-Tx1 and L-Tx2, and the single-head attention*

$$\mathrm{ATTN}_l(Z) = Z + W_{VO}ZA_l(Z), \ A_l(Z) = \mathrm{softmax}(M + Z^T W_{QK} Z/\sqrt{d}),$$
$$\text{or } \mathrm{ATTN}_l(Z) = Z + W_O W_V Z A_l(Z), \ A_l(Z) = \mathrm{softmax}(M + Z^T W_K^T W_Q Z/\sqrt{d}), \tag{6}$$

*respectively for the architecture L-T1x and L-T2x. The matrix* $M$ *is the masking matrix whose entries are* $-\infty$ *on the lower triangle and* $0$ *on the upper triangle and the diagonal.*

**Remark 3.3.** *Both of the above definitions consider the post-LN versions of ResNets and transformers, where the LayerNorm acts in between residual connections. We work with this version here because the arguments are cleaner, but the results do not qualitatively change if we used pre-norm ResNets or transformers instead. We discuss pre-LN architectures and their proof in Appendix B.*

**Neural collapse metrics and generalized unconstrained features model (GUFM).** Regardless of the model, let $h_\theta(\cdot)$ be the output of the corresponding architecture before the last layer, i.e., the feature on which neural collapse is defined. We denote by $x_{ki}^l$ the $i$-th sample of the $k$-th class in the $l$-th layer. We define $\mu_k^l := \frac{1}{n}\sum_{i=1}^n x_{ki}^l$ as the class-means in the $l$-th layer and $\mu_G^l := \frac{1}{K}\sum_{k=1}^K \mu_k^l$ as the global mean. Let $\Sigma_W^l := \frac{1}{N}\sum_{k,i=1}^{K,n}(x_{k,i}^l - \mu_k^l)(x_{k,i}^l - \mu_k^l)^T$ and $\Sigma_B^l := \frac{1}{K}\sum_{k=1}^K (\mu_k^l - \mu_G^l)(\mu_k^l - \mu_G^l)^T$ be the within- and between-class variability matrices in the $l$-th layer, and $M^l$ be the matrix of class-means stacked column-wise. Let $E_K = I_K - \mathbf{1}_K \mathbf{1}_K^T$ be the un-rotated ETF matrix. We define below neural collapse and its metrics for generic matrices.

**Definition 3.4.** *Any pair* $(W, X)$ *of matrices s.t.* $W$ *has at least as many columns as rows,* $X$ *has* $N = Kn$ *columns and they can multiply as* $WX$ *has the following NC metrics:*

- $\mathrm{NC1}(W, X) = \frac{\mathrm{tr}(\Sigma_W)}{\mathrm{tr}(\Sigma_B)}$*, i.e., the ratio of within- and between-class variability.*

- $\mathrm{NC2A}(W, X) = \frac{\min_{c \geq 0}\left\| WW^T - cE_K \right\|_F}{\|WW^T\|_F}$*, i.e. the distance of* $WW^T$ *from the closest (scaled) ETF.*

- $\mathrm{NC2B}(W, X) = \frac{\min_{c \geq 0}\left\| WW^T - cI_K \right\|_F}{\|WW^T\|_F}$*, i.e. the distance of* $WW^T$ *from the closest (scaled) identity.*

- $\mathrm{NC3}(W, X) = 1 - \frac{1}{N}\sum_{k,i=1}^{K,n}\cos(x_{ki}, W_{k:})$*, i.e., one minus the average cosine similarity between the samples and the corresponding row of* $W$*.*

A model is said to exhibit NC if all metrics are $0$ and approximate NC if all metrics are close to zero. NC2A is defined for CE loss or MSE loss with unregularized bias in the last layer, and NC2B is defined for MSE loss with bias-free last layer.

We consider the following optimization problem:

$$\min_\theta \ \mathcal{L}(f_\theta(X), Y) + \frac{\lambda}{2}\left\| \bar\theta \right\|^2, \tag{7}$$

where $\lambda > 0$, $\bar\theta$ is the subset of parameters that excludes biases and the parameters in embedding layers $(W_e, W_p, W_0)$, and $\mathcal{L}$ is a continuous, non-negative loss. Let $\mathcal{L}_{\mathrm{CE}}, \mathcal{L}_{\mathrm{MSE}}$ be CE and MSE loss, and $\mathcal{L}_{L,m}(\theta)$ be the loss of the $L$-RN$m$, $L$-T1$m$, or $L$-T2$m$ architecture with parameters $\theta$ (it will

be clear from the context whether this refers to a ResNet or a transformer). We denote by $\mathcal{L}^*_{L,m}$ the optimal such loss value and by $\mathcal{M}^{L,m}_\epsilon := \{\theta : \mathcal{L}_{L,m}(\theta) \leq \mathcal{L}^*_{L,m} + \epsilon\}$ the set of parameters $\epsilon$-close to the optimum. We denote by $\tilde{\mathcal{M}}_L$ the set of all pairs $(W_L, h_\theta(X))$ s.t. $\theta$ (including $W_L$) is in $\mathcal{M}^{L,1}_0$. In our theoretical analysis, we will reduce the end-to-end problem (7) into a simpler unconstrained features model, which we define below.

**Definition 3.5.** *Given a continuous loss $\mathcal{L} \geq 0$ and an equivalence relation $\mathcal{R}$ on $\{1, \ldots, N\}$, the generalized unconstrained features model (GUFM) refers to the following optimization problem:*

$$\min_{W,X} \; \mathcal{L}(WX, Y) + \frac{\lambda}{2} \|W\|^2_F, \tag{8}$$

$$s.t. \quad \|x_i\| = \sqrt{d}, \quad x_i^T \mathbf{1}_d = 0, \quad for \; i \in \{1, \ldots, N\},$$

$$x_i = x_j, \quad for \; i, j \in \{1, \ldots, N\}, \quad i \sim_\mathcal{R} j,$$

*where $W \in \mathbb{R}^{K \times d}, X = [x_1, \ldots, x_N] \in \mathbb{R}^{d \times N}$ and $Y \in \mathbb{R}^{K \times N}$. Let $\mathcal{L}_{GUFM}(W, X)$ be the loss of the feasible pair $(W, X)$ under this model, $\mathcal{L}^*_{GUFM}$ the optimal such loss and $\mathcal{M}^{GUFM}_\epsilon := \{(W, X) \in \mathcal{M} : \mathcal{L}_{GUFM}(W, X) \leq \mathcal{L}^*_{GUFM} + \epsilon\}$, with $\mathcal{M}$ the set of feasible solutions.*

The mean-zero constraint $x_i^T \mathbf{1}_d = 0$ comes from the application of this model to ResNets and transformers having LayerNorm before the last layer, which allows the model to represent only zero-mean solutions. We note that this is without loss of generality for CE/MSE loss, since for those losses the optimum is zero-mean. The equivalence relation constraints are introduced to account for potential hard constraints from the input data where we may have identical samples or contexts that may or may not be in the same class. Again, for CE/MSE loss this is without loss of generality, given that all identical contexts are always labeled with the same class (see Assumption 4.4).

## 4 Main results

### 4.1 Analysis of the generalized unconstrained features model

We start with a lemma showing that nearly-optimal solutions of the GUFM problem above must necessarily be close to the global optima.

**Lemma 4.1.** *Denote as $\mathrm{distmax}(A, B) = \sup_{x \in A} \mathrm{dist}(x, B)$ for any sets $A, B$. Then, we have*

$$\limsup_{\epsilon \to 0} \; \mathrm{distmax}(\mathcal{M}^{GUFM}_\epsilon \backslash \mathcal{M}^{GUFM}_0, \mathcal{M}^{GUFM}_0) = 0. \tag{9}$$

*Proof.* Assume by contradiction there exists a sequence $(X_n, W_n)^\infty_{n=1}$ of points such that $\lim_{n \to \infty} \mathcal{L}_{\mathrm{GUFM}}(W_n, X_n) = \mathcal{L}^*_{\mathrm{GUFM}}$ but $\limsup_{n \to \infty} \mathrm{dist}((W_n, X_n), \mathcal{M}^{\mathrm{GUFM}}_0) = c > 0$. Then, since the feasible set of GUFM is compact (for $W$, take a large-enough ball around 0 that must contain the global optimum), we can choose a subsequence $(X_{n_k}, W_{n_k})^\infty_{k=1}$ having an accumulation point $(\bar{W}, \bar{X})$ in the feasible set and s.t. $\mathrm{dist}((\bar{W}, \bar{X}), \mathcal{M}^{\mathrm{GUFM}}_0) > 0$ (first picking a subsequence for which the limsup above is realized and only choosing a subsequence with accumulation point from this subsequence; then using the continuity of the distance to conclude). From the continuity of the loss function, it must follow $\mathcal{L}_{\mathrm{GUFM}}(\bar{W}, \bar{X}) = \mathcal{L}^*_{\mathrm{GUFM}}$, which also implies $(\bar{W}, \bar{X}) \in \mathcal{M}^{\mathrm{GUFM}}_0$. However, this is a contradiction because the distance of this point from $\mathcal{M}^{\mathrm{GUFM}}_0$ is both 0 and bigger than 0. $\square$

Next, we focus on CE and MSE loss, showing that the optima of the corresponding GUFMs (denoted by UFM-CE and UFM-MSE) exhibit NC.

**Lemma 4.2.** *Assume that only the samples within the same class are in relation $\mathcal{R}$. Then, the global optima $\mathcal{M}^{UFM\text{-}CE}_0$ and $\mathcal{M}^{UFM\text{-}MSE}_0$ are all perfectly collapsed, i.e., for all $(W, X) \in \mathcal{M}^{UFM\text{-}CE}_0$, $\mathrm{NC1}(W, X) = \mathrm{NC2A}(W, X) = \mathrm{NC3}(W, X) = 0$ and for all $(W, X) \in \mathcal{M}^{UFM\text{-}MSE}_0$, $\mathrm{NC1}(W, X) = \mathrm{NC2B}(W, X) = \mathrm{NC3}(W, X) = 0$. Conversely, for any feasible pair $(W, X)$ s.t. $\mathrm{NC1}(W, X) = \mathrm{NC2A}(W, X) = \mathrm{NC3}(W, X) = 0$, there exists a unique scalar $c$ s.t. $(cW, X) \in \mathcal{M}^{UFM\text{-}CE}_0$; and for any feasible pair $(W, X)$ s.t. $\mathrm{NC1}(W, X) = \mathrm{NC2B}(W, X) = \mathrm{NC3}(W, X) = 0$, there exists a unique scalar $c$ s.t. $(cW, X) \in \mathcal{M}^{UFM\text{-}MSE}_0$.*

The proof is deferred to Appendix A. For the CE loss, it is based on an adaptation of the results in [71]. For the MSE loss, we compute the global optima by lower-bounding the loss, solving the problem for the lower-bound and showing that the loss and its lower-bound agree at these optima.

## 4.2 Deep single-layer architectures are collapsed at the global optimum

We first consider ResNets/transformers with one linear layer per MLP block.

**Theorem 4.3.** *Let the architecture be L-RN1 or L-Tx1 for $x \in \{1, 2\}$. Assume the inner dimension of the L-Tx1 is at least $2V + 2$ and the inner dimension of L-RN1 is at least 4. Consider the optimization problem (7) with $\lambda$ independent of the number of layers. Consider also its corresponding GUFM (8) with the same loss $\mathcal{L}$ and the equivalence relation defined by pairs of samples in $X$ that coincide (for transformers, these correspond to a pair of identical contexts). If $\mathcal{L}^*_{GUFM} > 0$, then*

$$\limsup_{L \to \infty} \ \text{distmax}(\tilde{\mathcal{M}}_L \backslash \mathcal{M}_0^{GUFM}, \mathcal{M}_0^{GUFM}) = 0. \tag{10}$$

The result above provides a reduction of the end-to-end training objective of a deep-enough architecture to a GUFM using the same loss. This has two important implications. First, it shows that optimal deep ResNets and transformers can represent the optimal solution of the corresponding GUFM problem. As formalized in Corollary 4.5, this gives a precise characterization of the structure of feature representations in the last layer at the global optimum – the first result of this sort for modern architectures beyond MLPs. Second, it provides a theoretical justification for using the UFM to explain the emergence of NC, showing that the UFM does not oversimplify the problem even when dealing with ResNets and transformers. We note that the lower bound of $2V + 2$ on the dimension of transformers is for technical convenience, and it can be loosened to a lower bound that does not depend on $V$. We now give a proof sketch deferring the complete argument to Appendix A.

*Proof sketch.* We start with the sketch for the $L$-RN1 model. Notice that $\mathcal{L}_{L,1}(\theta) = \mathcal{L}_{\text{GUFM}}(W_L, X_L) + \frac{\lambda}{2} \sum_{l=1}^{L-1} \|W_l\|_F^2$. The goal is to show $\mathcal{L}^*_{\text{GUFM}} = \lim_{L \to \infty} \mathcal{L}^*_{L,1}$, which implies that $\mathcal{L}_{\text{GUFM}}(W_L, X_L)$ for $(W_L, X_L) \in \tilde{\mathcal{M}}_L$ converges to $\mathcal{L}^*_{\text{GUFM}}$. Thus, $(W_L, X_L) \in \tilde{\mathcal{M}}_L$ must also belong to $\mathcal{M}_\epsilon^{\text{GUFM}}$ for $\epsilon$ small enough, which by Lemma 4.1 guarantees the convergence as in (10).

Note that we can represent a one-block-deeper ResNet that perfectly copies the original ResNet by simply adding an identity block with zero weight matrices/biases and residual connection left untouched. Thus, $\mathcal{L}^*_{L,1}$ is non-increasing in $L$ and it suffices to prove the limit for any sequence of $L$'s going to infinity. We will prove it by explicitly constructing a sequence of $L$-RN1 ResNets s.t. their losses converge to $\mathcal{L}^*_{\text{GUFM}}$ as $L \to \infty$. This crucially relies on the fact that it is possible to almost perfectly fit the training data $X_0$ with ResNets so that the sum of Frobenius norms of all their layers converges to 0. This is a special property of residual networks that qualitatively differs from MLPs.

To build the intuition on why this is possible, consider a 1D example where we want to fit the label $\exp(a)$ when the input is 1 with a 1D ResNet. Let $x$ be a shared weight across all layers. Then, we need $\exp(a) = (1 + x)^L$, which can be asymptotically achieved by setting $x = a/L$. Importantly, the sum of Frobenius norms $\sum_{l=1}^L \left(\frac{a}{L}\right)^2 = \frac{a^2}{L}$ vanishes as $L \to \infty$. In other words, by "splitting" the mapping done by a single ResNet layer into $L$ layers with smaller weights, the total cost is smaller. A similar intuition was also used in [4].

For a multi-dimensional ResNet and general data, the idea of the construction is to split the blocks of the ResNet into $N$ groups, with each group moving only a single sample (for simplicity we assume all samples are distinct in this proof sketch). In this way, it is possible to split the layers within one group into several layers implementing the same mapping with a smaller Frobenius norm. Each sample has a predefined smooth trajectory from its initial position to the near-optimal position under the GUFM, and the group of blocks responsible for moving this sample approximates a smooth movement along this trajectory. As the depth increases, the total cost of these layers decreases, since each of them gets smaller with rate $1/L$, as in the 1D computation above, thus giving the desired result.

Next, let us consider the $L$-Tx1 model. The key observation is that transformers are basically a strict extension of ResNets, with attention layers being the only extra component. However, setting the attention layers to 0 and directly applying the result above for the $L$-RN1 model does not immediately work. In fact, for each token, we need attention layers to acquire information from previous tokens, which may be useful to fit the label. At the same time, if we want the sum of Frobenius norms of all

the layers to converge to zero, so must all the key and query matrices, which makes attention scores in all layers necessarily converge to uniform across the entire past.

The solution is to design the embedding layer and the first transformer block so that distinct contexts of $X_0$ remain distinct and their distances do not converge to $0$ too fast (this would disrupt our construction for ResNets which implicitly assumes that the initial distances between samples are constant w.r.t. $L$). The embedding layer and first attention layer encode the contexts so that the $j$-th entry contains the history (encoded in binary) of all the tokens belonging to the $j$-th class from the past. At this point, a slight adjustment of the construction for ResNets finishes the proof. $\qquad\square$

Note that the uniform attention which we use in the proof, although asymptotically optimal, is not expected to be optimal for any moderate number of layers and we only use it in finite-layer constructions for mathematical convenience and consistency with the asymptotic case. We highlight that Theorem 4.3 holds for any continuous loss. By considering CE or MSE for which the global optima of the corresponding GUFMs are collapsed by Lemma 4.2, the emergence of collapse in ResNets and transformers is readily obtained, assuming the following about training data:

**Assumption 4.4.** *For the ResNet architecture, we assume all training samples in $X$ to be unique. For the transformer architecture, we assume the labels $Y$ to be uniquely determined by the context, i.e., two identical contexts in two different input sequences will be assigned the same label.*

**Corollary 4.5.** *Let the architecture be L-RN1 or L-Tx1 for $x \in \{1, 2\}$. Assume the training data $(X, Y)$ satisfies Assumption 4.4 and all the assumptions of Theorem 4.3. Using CE or MSE loss, all global optima of the optimization problem (7) exhibit approximate neural collapse which gets tighter as $L$ increases.*

We make several remarks about this result.

**Rate of convergence.** While the results are stated asymptotically for simplicity, one can readily recover a convergence rate of the global optimum to NC from the argument. In particular, since the total regularization of the layers scales as $L^{-1}$, the global optima can only be suboptimal w.r.t. the GUFM objective with the same scaling. Then, assuming a differentiable loss (e.g., CE or MSE), the distance from the optima scales as the inverse of the power in the Taylor approximation of the loss at the global optima in the flattest direction, up to logarithmic factors that come from making a finer approximation. Now, for the CE loss, the leading term is quadratic: by using the chain rule, the slope of CE at the optimum is non-zero, and the sum of exponentials of dot-products between $X, W$ is quadratic as we approach the ETF. Thus, the convergence in distance is $\tilde{O}(L^{-1/2})$, where $\tilde{O}$ omits logarithmic factors. For the MSE loss we compute by error analysis in the proof of Lemma 4.2 that the convergence rate is also $\tilde{O}(L^{-1/2})$.

**Language modeling.** When considering the transformer architecture, we require the labels to be unique given a specific context. While this is a realistic assumption in vision or language classification tasks (e.g., sentiment analysis, harmful content classification, spam detection), it does not apply to language pretraining, where a single context may have many different continuations. In fact, in the setting of non-unique continuations, neural collapse is *not* to be expected, and the optimal structure was discussed [50, 66] by using a form of UFM. We remark that Theorem 4.3 shows that the optimal solutions identified in these works are exhibited by transformers, as long as they satisfy the conditions in (8). This is the case, for instance, in some symmetric settings, see Proposition 2 in [66] with a slight modification in the underlying UFM (the authors consider weight decay instead of norm constraints on the features), where the optimal limiting solution is indeed collapse. In non-symmetric cases, while NC is not expected to be optimal (as in the case with class imbalance [51]), transformers still represent the optimal zero-mean solution of the underlying UFM, whatever that is. This allows future work to focus on solving the application-relevant UFM in the corresponding setting and then use Theorem 4.3 to conclude that the solutions are globally optimal end-to-end.

**Deep neural collapse.** Although our theory focuses on last-layer geometry, the analysis sheds some light on the collapse in the earlier layers as well. In particular, one can readily obtain that any finite number of layers at the end of the network converges to neural collapse (with the exception of NC3 which has a different formulation for multi-layer collapse). Note that adding a residual connection (as in ResNets and transformers) resolves the inconsistency of deep UFMs pointed out in [49], where it is shown that the global optima of the deep UFM in the multi-class setting do *not* exhibit neural collapse. In fact, the optimal solution of a deep UFM with residual connections is obtained by simply copying the shallow UFM in the first layer and setting all remaining layers to 0. We also remark

that, from the argument of Theorem 4.3, it follows that the global optima of the last $\tilde{L}$ layers of the network ($\tilde{L}$ being a constant independent of $L$) converge to the global optima of the corresponding deep GUFM with residual connections and depth $\tilde{L}$.

In contrast, understanding the emergence of neural collapse for a small, but constant fraction of the final layers of the network appears to require a different approach. Intuitively, if the network starts processing all samples at once from some layer onwards (which is expected to improve the loss w.r.t. our construction), then the collapse is progressive and occurs to some extent already in a constant fraction of the final layers, see also the discussion in [58].

### 4.3 Deep double-layer architectures are collapsed at the global optimum with vanishing regularization

Let us now consider ResNets with two linear layers per block and transformers with two linear layers per MLP sub-block (the number of matrices in the attention sub-block does not affect the result). Then, we show that neural collapse is globally optimal, provided that the regularization strength in all layers except the last one decreases with the depth $L$.

**Theorem 4.6.** *Let the architecture be L-RN2 or L-Tx2 for $x \in \{1, 2\}$. Assume the inner dimension of the L-Tx1 is at least $2V + 2$ and the inner dimension of L-RN1 is at least 4. Consider the optimization problem*

$$\min_{\theta} \ \mathcal{L}(f_\theta(X), Y) + \frac{\lambda_L}{2} \|W_L\|_F^2 + \frac{\lambda(L)}{2} \|\bar{\theta}\|^2, \tag{11}$$

*where $\lambda_L$ is a regularization on the weight matrix of the last layer that does not depend on $L$ and $\lambda(L)$ is a depth-dependent regularization s.t. $\lambda(L) = o(\log(L)^{-1})$. Consider the corresponding GUFM with regularization $\lambda_L$. If $\mathcal{L}_{GUFM}^* > 0$, then*

$$\limsup_{L \to \infty} \ \text{distmax}(\tilde{\mathcal{M}}_L \backslash \mathcal{M}_0^{GUFM}, \mathcal{M}_0^{GUFM}) = 0. \tag{12}$$

The reason why the regularization is required to be vanishing can be already seen from the 1D example mentioned in the proof sketch of Theorem 4.3: in order to ensure that $(1 + x^2)^L$ converges to $\exp(a)$ as $L \to \infty$, one needs to pick $x = \frac{\sqrt{a}}{\sqrt{L}}$, which implies that the sum of squares $\sum_{l=1}^{L} \left(\frac{\sqrt{a}}{\sqrt{L}}\right)^2$ is of constant order w.r.t. $L$. In fact, the requirement on vanishing regularization is necessary for the statement to be true, and the result also cannot hold if both $\lambda(L)$ and $\lambda_L$ are vanishing. An additional discussion on this point, together with a concrete dataset for which collapse cannot be reached, are provided in Appendix C. Understanding the structure of the optimal representations for double-layer architectures in the regime of constant regularization represents an exciting future direction.

The proof of Theorem 4.6 is similar to that of Theorem 4.3. In particular, the first layers of the blocks are defined in the same way, and the second layers are set to act as a projection matrix on the space spanned by the output of the first layer, which has rank 1. Furthermore, the scalings of these layers are split in identical square roots of the scaling of the original layer. Thus, the sum over the squared Frobenius norms is constant w.r.t. $L$, which requires $\lambda(L)$ to vanish in $L$. The detailed proof is deferred to Appendix A. We conclude the section by stating the approximate optimality of NC in the global optima of double-layer architectures under CE or MSE loss.

**Corollary 4.7.** *Let the architecture be L-RN2 or L-Tx2 for $x \in \{1, 2\}$. Assume the training data $(X, Y)$ satisfies Assumption 4.4 and the assumptions of Theorem 4.6. Using CE or MSE loss, all global optima of the optimization problem (11) exhibit approximate neural collapse which gets tighter as $L$ increases.*

## 5 Experimental results

Our theoretical results suggest an improvement of the NC metrics at the global optima as the depth increases. To empirically verify whether this effect is already present at moderate depths and for solutions found by gradient descent, we train ResNets and transformers on MNIST [29], CIFAR10 [31] and IMDB [21] with increasing depths in $\{2, 3, 5, 8, 13, 21, 34\}$. The hidden dimension is 64, the learning rate 0.005 for vision and 0.001 for language and the (constant) regularization 0.005 for architectures having one linear layer per block and $0.005/L$ for architectures having two linear layers

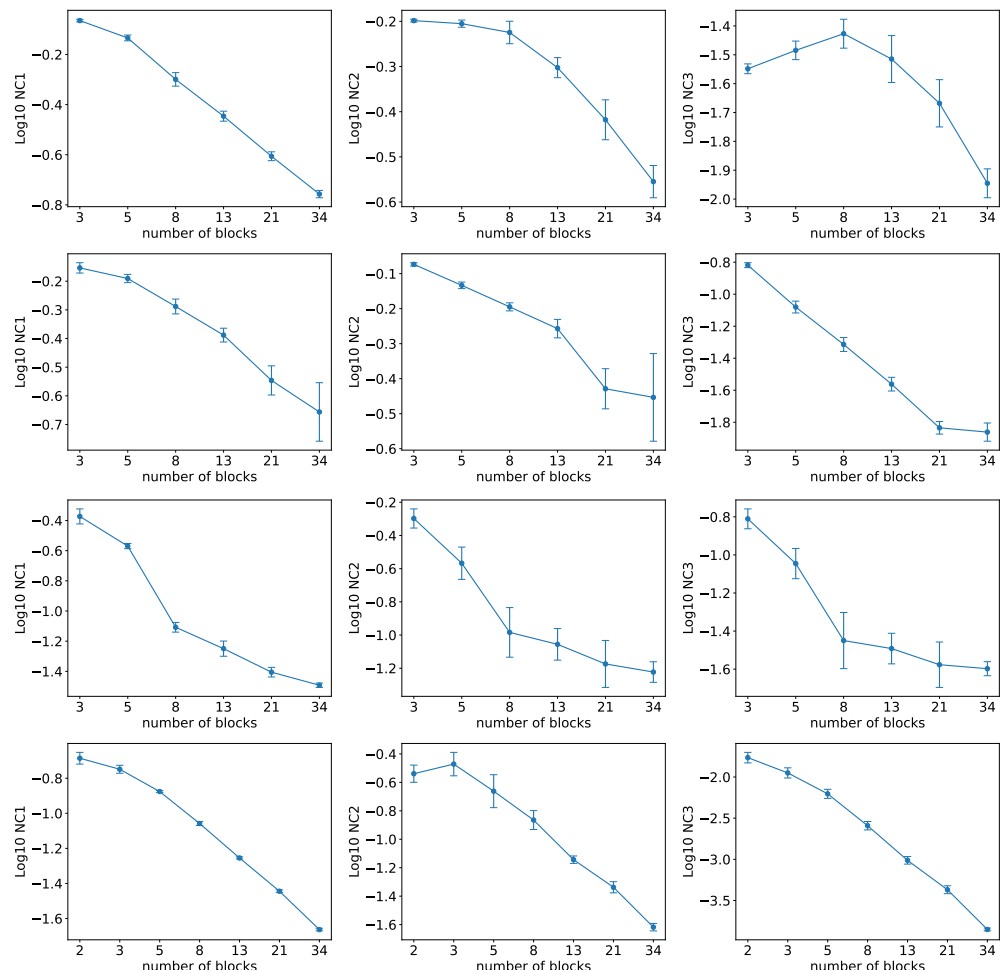

Figure 1: $\log_{10}$ of NC1, NC2 and NC3 metrics respectively in the left, middle and right column, as a function of the number of blocks $L$. *First row*: $L$-RN1 on CIFAR10; *second row:* $L$-T11 on CIFAR10; *third row:* pre-LN $L$-T11 on IMDB; *Fourth row:* $L$-RN2 on MNIST with $\lambda \propto L^{-1}$.

per block. Each setting is trained for 5 different random seeds for 5000 epochs on CE loss, the results are averaged, and the error bars at one standard deviation are reported. We use pre-LN transformers for language experiments and, due to training instabilities, only report the runs which converged by the end of the training. See Appendix D for additional experimental results.

Figure 1 shows the three NC metrics at convergence, as a function of the depth of the architecture. The results are in agreement with the theory developed in Section 4: across different datasets and architectures, NC metrics improve with depth, even when the solutions are obtained via gradient descent. Furthermore, for large enough depth, the plots roughly follow a log-linear trend with an average slope of around $-0.335$, especially for ResNets. This suggests a polynomial dependence between NC metrics and depth $L$, which is also consistent with our theory, see the remark on the rate of convergence in Section 4.2, including the quantitative estimate of the slope being between $-1/2$ and $-1/4$. The metrics are generally a bit larger (meaning less strong collapse) than the ones usually measured in MLPs [48], but this is because collapse in ResNets is approached more slowly due to different loss landscape. Finally, we remark that [17, 45] consider the effect of depth, but instead focus on the progression of NC metrics across layers, rather than evaluating such metrics in the last layer as a function of the overall depth.

## 6 Conclusion

This work provides global optimality guarantees for neural collapse in two modern architectures: ResNets and transformers. Besides [23] for simplified MLPs, this is the first end-to-end global optimality result for NC in deep networks. Our approach involves a reduction to a general form of unconstrained features model that holds for any continuous loss. This provides a formal justification for the validity of the UFM as a modeling principle and it motivates future work on it in new settings, such as language modeling [50, 66]. Experimental results confirm our theoretical predictions on standard datasets trained via gradient descent, thus providing a simple recipe for practitioners thriving to achieve a strong collapse in applications [34, 38]: just increase the depth.

Although the analysis covers a wide range of models, the behavior of global optima for architectures with two linear layers per block and constant regularization remains open. While we know that NC is not asymptotically reached for all datasets, studying the tradeoff between representation cost and fit loss (and, thus, the extent of NC in global optima) is an important open problem. Beyond that, our work suggests several interesting future directions. First, by improving the constructions used to prove Theorems 4.3 and 4.6, one could obtain more refined bounds on the convergence rate in terms of the depth $L$, leading to sharp NC guarantees already for a moderate number of layers. Second, it would be very exciting to adjust our results to describe deep neural collapse and quantify the evolution of NC metrics across depth, thereby refining the results in [58].

## Acknowledgements

M. M. and P. S. are funded by the European Union (ERC, INF$^2$, project number 101161364). Views and opinions expressed are however those of the author(s) only and do not necessarily reflect those of the European Union or the European Research Council Executive Agency. Neither the European Union nor the granting authority can be held responsible for them. This research was supported by the Scientific Service Units (SSU) of ISTA through resources provided by Scientific Computing (SciComp).

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

## A  Deferred proofs

**Lemma 4.2.** *Assume that only the samples within the same class are in relation $\mathcal{R}$. Then, the global optima $\mathcal{M}_0^{UFM\text{-}CE}$ and $\mathcal{M}_0^{UFM\text{-}MSE}$ are all perfectly collapsed, i.e., for all $(W, X) \in \mathcal{M}_0^{UFM\text{-}CE}$, $NC1(W, X) = NC2A(W, X) = NC3(W, X) = 0$ and for all $(W, X) \in \mathcal{M}_0^{UFM\text{-}MSE}$, $NC1(W, X) = NC2B(W, X) = NC3(W, X) = 0$. Conversely, for any feasible pair $(W, X)$ s.t. $NC1(W, X) = NC2A(W, X) = NC3(W, X) = 0$, there exists a unique scalar $c$ s.t. $(cW, X) \in \mathcal{M}_0^{UFM\text{-}CE}$; and for any feasible pair $(W, X)$ s.t. $NC1(W, X) = NC2B(W, X) = NC3(W, X) = 0$, there exists a unique scalar $c$ s.t. $(cW, X) \in \mathcal{M}_0^{UFM\text{-}MSE}$.*

*Proof.* For both losses, we will relax the problem and ignore the constraints coming from the equivalence relation $\mathcal{R}$. Then, we prove that NC1 holds in all of these cases, which grants equivalence between the relaxed and original problem.

For the CE loss, we apply Theorem 3.1 of [71]. In particular, from this theorem it follows that the optimal solutions of the regularized UFM-CE (not a-priori equivalent to (8) because of the feature constraint) exhibit neural collapse. From their proof, it is also clear that not only does the ratio between the sizes of the optimal $w_k$ and $x_{ki}$ only depend on the ratio of the regularization terms, but also that the absolute size of these vectors is an increasing function of the regularization strength, with the limit as $\lambda \to \infty$ being infinity. Therefore, let us pick $\lambda_W$ from the paper to be $\lambda$ in (8), while we find $\lambda_H$ s.t. the optimal solutions of the problem in [71] have norm $\sqrt{d}$. Then, the global optima of the regularized UFM-CE are exactly those of the UFM-CE we consider in (8).

To see the last statement, assume by contradiction that there is a global optimum of the problem in (8) which is not a global optimum of the regularized UFM-CE. Then, we can plug this solution into the regularized UFM-CE. Since it is not a global optimum, there exists a solution with strictly lower loss, and this optimum is guaranteed to have unit norm features. By plugging this optimum into (8), we must obtain a loss that is better than the optimal one, since the objectives are equivalent in this case. This leads to a contradiction. Similar arguments give that there cannot exist a global optimum of the regularized UFM-CE which is not a global optimum of (8), thus proving the desired equivalence.

For the MSE loss, we perform a direct computation which includes a perturbation analysis. To simplify the loss landscape, we start by defining a lower bound on the UFM-MSE loss, which we will analyze first. Denote

$$\underline{\mathcal{L}}_{\text{UFM-MSE}} := \frac{1}{2N} \sum_{k,i=1}^{K,n} (w_k^T x_{ki} - 1)^2 + \frac{\lambda}{2} \|W\|_F^2 \tag{13}$$

and $\underline{\mathcal{M}}_\epsilon^{\text{UFM-MSE}}$ the corresponding near-optimal set. Note that (13) is separable in the index $k$, thus we are facing $K$ identical, independent optimization problems. We will now do a series of partial conditional optimizations and comment on the cost of deviating from these conditional optima. First, conditioning on any $w_k$ (corresponding to the $k$-th row of $W$), we can almost exactly specify the optimal values of $x_{ki}$ for any $i$. In particular, if $\|w_k\| \leq d^{-\frac{1}{2}}$, then the optimal solution is $x_{ki} = \sqrt{d} \cdot w_k / \|w_k\|$. If $\|w_k\| > d^{-\frac{1}{2}}$, then the optimal solution is any vector on a hypersphere such that $w_k^T x_{ki} = 1$. In the former case, for each $x_{ki}$, a deviation from the optimal value of the dot-product $w_k^T x_{ki} = \sqrt{d} \|w_k\|$ results in a quadratic increase in the loss around the optimal point (the cosine function has zero linear term in the Taylor expansion and non-zero quadratic term) or quartic if $\|w_k\| = d^{-\frac{1}{2}}$ (because the loss at optimum would be 0 and being itself a quadratic function, the effects would multiply). In the case $\|w_k\| > d^{-\frac{1}{2}}$, the loss increase around the optimum is again quadratic. Therefore, in all cases the maximum allowed deviation from the optimum given an extra loss of $\epsilon$ is at most $\mathcal{O}(\epsilon^{1/4})$ and, thus, goes to 0 as $\epsilon$ goes to zero.

Now, denote $z \equiv \|w_k\|$. The loss of the $k$-th group only depends on $z$ and $x_{ki}$, but plugging-in the optimal value after solving for $x_{ki}$ we arrive at a single-dimensional objective that only depends on $z$:

$$\frac{1}{2K}(1 - z\sqrt{d}) \max(1 - z\sqrt{d}, 0) + \frac{\lambda}{2} z^2.$$

From the form of this optimization problem, it is clear that the unique global optimum is reached on $(0, d^{-\frac{1}{2}})$. The solution is simply $\frac{1}{\sqrt{d}(1+\lambda K)}$. First, we note that, for fixed $\lambda, K$, this solution is strictly

smaller than $d^{-\frac{1}{2}}$ with non-zero margin. Second, any deviation from this optimal solution will result in a quadratic increase in the loss function, therefore for a fixed extra loss of $\epsilon$, the maximum allowed deviation of $\|z_k\|$ from its optimal value is $\mathcal{O}(\epsilon^{1/2})$, which also goes to 0 with $\epsilon$ going to 0. Moreover, since its optimal value (and also maximum allowed deviation for $\epsilon$ small-enough) is strictly smaller than $d^{-\frac{1}{2}}$, we know that the optimal value of the $x_{ki}$ is indeed $\sqrt{d} \cdot w_k / \|w_k\|$ and the maximum allowed deviation is also $\mathcal{O}(\epsilon^{1/2})$.

The function value in (13) cannot be optimized any further, thus we know what $\underline{\mathcal{M}}_0^{\text{UFM-MSE}}$ is. In particular, the solutions in $\underline{\mathcal{M}}_0^{\text{UFM-MSE}}$ must satisfy the NC1 and NC3 properties. Now, if the global optima of (8) with MSE loss and (13) are equal, then $\mathcal{M}_0^{\text{UFM-MSE}} \subset \underline{\mathcal{M}}_0^{\text{UFM-MSE}}$ and thus the optimal solutions of (8) with MSE loss must also satisfy the NC1 and NC3 criteria from the lemma statement.

To show that the global optima are equal and to argue about NC2, we turn back to the original problem (8) with MSE loss. Since we know that the optimal solutions agree, we can focus directly on $\underline{\mathcal{M}}_0^{\text{UFM-MSE}}$. After plugging any optimal solution of (13) into $\mathcal{L}_{\text{UFM-MSE}}$, we see that the regularization part is constant, so we are left with optimizing the fit part. Analyzing the loss incurred by $x_{ki}$ on position $l \neq k$ we see that it is $(w_l^T x_{ki})^2 = (w_l^T w_k)^2 d(1 + \lambda K)^2$. Summing this over all indices and samples (using the symmetries) we see that the total loss is proportional to the Frobenius norm of the off-diagonal elements of $WW^T$. Therefore, a lower-bound on the loss is 0, which is achievable provided $W$ has at least as many columns as rows, as assumed in the lemma. Let us simply choose $W$ to be a scaled orthogonal matrix, and note that the loss cannot be optimized any further. Thus, we see that $\mathcal{L}_{\text{UFM-MSE}}^* = \underline{\mathcal{L}}_{\text{UFM-MSE}}^*$ and the solutions of (8) with MSE must satisfy NC2. Any deviation of $W$ from an orthogonal matrix will result in an increase in the loss which is at least quartic: given a fixed extra loss of $\epsilon$, the solution in $\mathcal{M}_\epsilon^{\text{UFM-MSE}}$ must be $\mathcal{O}(\epsilon^{1/4})$ close to an orthogonal matrix.

Finally, the converse statements also readily follow from the above computations. □

**Theorem 4.3.** *Let the architecture be L-RN1 or L-Tx1 for $x \in \{1, 2\}$. Assume the inner dimension of the L-Tx1 is at least $2V + 2$ and the inner dimension of L-RN1 is at least 4. Consider the optimization problem (7) with $\lambda$ independent of the number of layers. Consider also its corresponding GUFM (8) with the same loss $\mathcal{L}$ and the equivalence relation defined by pairs of samples in $X$ that coincide (for transformers, these correspond to a pair of identical contexts). If $\mathcal{L}_{GUFM}^* > 0$, then*

$$\limsup_{L \to \infty} \text{distmax}(\tilde{\mathcal{M}}_L \setminus \mathcal{M}_0^{GUFM}, \mathcal{M}_0^{GUFM}) = 0. \tag{10}$$

*Proof.* We first discuss how to deal with the equivalence relation $\mathcal{R}$. The argument is identical whether we take individual samples if all samples are distinct, or we treat the equivalence classes as individual samples. Thus, for simplicity of notation we assume, without loss of generality, that the samples are all distinct.

We start with the proof for the L-RN1 model. Notice that $\mathcal{L}_{L,1}(\theta) = \mathcal{L}_{\text{GUFM}}(W_L, X_L) + \frac{\lambda}{2} \sum_{l=1}^{L-1} \|W_l\|_F^2$. The goal is to show $\mathcal{L}_{\text{GUFM}}^* = \lim_{L \to \infty} \mathcal{L}_{L,1}^*$. In that case, $\mathcal{L}_{\text{GUFM}}(W_L, X_L)$ must converge to $\mathcal{L}_{\text{GUFM}}^*$. Therefore, $(W_L, X_L)$ induced by $\theta \in \mathcal{M}_0^{L,1}$ must also belong to $\mathcal{M}_\epsilon^{\text{GUFM}}$ for $\epsilon$ arbitrarily small, which evoking Lemma 4.1 guarantees the convergence as defined in (10).

Note that we can represent a one-block-deeper ResNet that perfectly copies the original ResNet by simply adding an identity block with zero weight matrices/biases and residual connection left untouched. Thus, $\mathcal{L}_{L,1}^*$ is non-increasing in $L$ and it suffices to prove the limit for any sequence of $L$'s going to infinity. We will prove it by explicitly constructing a sequence of L-RN1 ResNets s.t. their losses converge to $\mathcal{L}_{\text{GUFM}}^*$ as $L \to \infty$.

Pick any $(W_L, X_L) \in \mathcal{M}_0^{\text{GUFM}}$ and relabel $H := X_L$. Thus, $h_{ki}$ is the feature representation of the sample $ki$ in the penultimate layer. Define $\bar{H}$ as the matrix of unique points $h_{ki}$, and let us index them with a single index as $\bar{h}_j$. Denote the number of these unique points as $\bar{K}$. If we write $j(ki)$ we mean the index $j$ such that $\bar{h}_j = h_{ki}$. Before starting the construction, we need to define a key data-dependent quantity. First, take $X_1 = \text{LN}(W_0 X_0 + b_0)$ for $b_0$ and $W_0$ sampled from a continuous distribution. Since points in $X_0$ are all disjoint, this property holds also for $X_1$ with probability 1. Moreover, with probability zero, any sample in $X_1$ is identical to $h_{ki}$ for any of the vectors in $H$. For simplicity, we will refer to $X_1$ and its samples as $X$ and drop the index. Fix an ordering of the points $x_{ki}$ as the lexicographical ordering of $(k, i)$. For each $(k, i)$ find a smooth oriented curve $\mathcal{G}_{ki}$

connecting $x_{ki}$ with $h_{ki}$ on the set of feasible points ($\sqrt{d}$ norm hypersphere with zero-sum entries) such that all of the following holds:

1. The curvature of $\mathcal{G}_{ki}$ defined as the Lipschitz constant of the unit-norm oriented tangent function $\mathcal{T}_{ki}$ is bounded by $B$.

2. For all $(l, j) > (k, i)$, $\max\limits_{x \in \mathcal{G}_{ki}} x_{lj}^T x \leq d(1 - m)$ for some $m > 0$, i.e., all the subsequent points $x_{lj}$ are far enough from the curve $\mathcal{G}_{ki}$.

3. There is precisely one point $\bar{x}_{ki} \in \mathcal{G}_{ki}$ such that $\bar{x}_{ki}^T h_{ki} = d(1 - cm)$, where $c > 1$ is chosen large enough. Denote $\bar{\mathcal{G}}_{ki}$ as the set of points on $\mathcal{G}_{ki}$ between $x_{ki}$ and $\bar{x}_{ki}$. Then, we assume that, for all $(l, j) < (k, i)$, $\max\limits_{x \in \bar{\mathcal{G}}_{k,i}, y \in \mathcal{G}_{l,j} \setminus \bar{\mathcal{G}}_{l,j}} x^T y \leq d(1 - m)$.

4. The length of $\mathcal{G}_{ki}$ is no more than $2\pi\sqrt{d}$.

5. $m$ is chosen small enough s.t. $10cm \leq (d - \max\limits_{j(ki) \neq j(lp)} \bar{h}_{j(ki)}^T \bar{h}_{j(lp)})/d$.

It is clear that a construction satisfying these properties exists, since the constants $B, c, m$ are chosen with respect to $X, H$ and the number of points we consider is finite. We also note that this requires the inner dimension of the representations to be at least 4 since this would not be possible on a 2D circle.

The idea of the construction is as follows. Take $L$ large enough and divide the layers into $N + \bar{K} + 1$ blocks. The first $N$ blocks are of the same number of layers $L_1$, and the depth of the last one will be specified later. Each of the first $N$ blocks of layers will focus on a single sample, while not changing the representation of the other samples at all. The goal of the $ki$-th block is to only move the $ki$-th sample on its curve towards $h_{ki}$, until it hits $\bar{x}_{ki}$. Then, the $\bar{K}$ next blocks of depth $L_2$ will move all the samples corresponding to the same $j(ki)$ at once, ever closer to their respective $\bar{h}_{j(ki)}$ vectors. Finally, the very last block which consists of the very last layer will simply be chosen as the optimal $W_L$ corresponding to $H$.

We will now construct explicitly all the layers. Denote by $W_{ki}^l, b_{ki}^l$ the parameters of the $l$-th layer of the $ki$-th block and define $x_{ki}^l$ to be the feature representation of the $ki$-th sample as an input to that layer. Consider a sphere with center $x_{ki}^l$ and radius $\frac{\alpha_{ki}^l m \sqrt{d}}{2\sqrt{d+m^2/4}}$, where $\alpha_{ki}^l$ is a small-enough number whose role will be clear soon. Since this sphere is small enough and $\mathcal{G}_{ki}$ has bounded curvature, there exists exactly one point $\tilde{x}_{ki}^{l+1}$ on the intersection between $\mathcal{G}_{ki}$ and the considered sphere which is closer to $h_{ki}$ as $x_{ki}^l$. Denote $d_{ki}^l = \frac{\tilde{x}_{ki}^{l+1} - x_{ki}^l}{\|\tilde{x}_{ki}^{l+1} - x_{ki}^l\|} = \frac{\tilde{x}_{ki}^{l+1} - x_{ki}^l}{\frac{\alpha_{ki}^l m \sqrt{d}}{2\sqrt{d+m^2/4}}}$. The weights are constructed as follows:

$$W_{ki}^l = \alpha_{ki}^l \frac{1 + \frac{m}{2} d_{ki}^l}{\sqrt{d + m^2/4}} \frac{(x_{ki}^l)^T}{\sqrt{d}}, \tag{14}$$

$$b_{ki}^l = -\left(1 - \frac{m}{2}\right) \frac{\alpha_{ki}^l \sqrt{d}}{\sqrt{d + m^2/4}} \mathbf{1},$$

if $(x_{ki}^l)^T h_{ki} \leq d(1 - cm)$, otherwise $W_{ki}^l = 0; b_{ki}^l = 0$. The $\alpha_{ki}^l$ is an optimizable parameter and since the form above is also $W$'s SVD, it is its singular value. Thus, $\sigma(W_{ki}^l x_{ki}^l + b_{ki}^l) = \frac{\alpha_{ki}^l m \sqrt{d}}{2\sqrt{d+m^2/4}} (\mathbf{1} + d_{ki}^l)$, while $\sigma(W_{ki}^l x_{st}^l + b_{ki}^l) = 0$ for any $(s, t) \neq (k, i)$ thanks to our margin definition. Therefore, before $x_{ki}$ hits its final destination, we have

$$x_{ki}^{l+1} = \text{LN}\left(x_{ki}^l + \frac{\alpha_{ki}^l m \sqrt{d}}{2\sqrt{d+m^2/4}}(\mathbf{1} + d_{ki}^l)\right) = \frac{\sqrt{d}\left(x_{ki}^l + \frac{\alpha_{ki}^l m \sqrt{d}}{2\sqrt{d+m^2/4}} d_{ki}^l\right)}{\left\|x_{ki}^l + \frac{\alpha_{ki}^l m \sqrt{d}}{2\sqrt{d+m^2/4}} d_{ki}^l\right\|} = \tilde{x}_{ki}^{l+1}.$$

From this, it is clear that $x_{ki}$ is moving along and on the curve, while the other samples stay stationary.

It remains to compute how fast $x_{ki}$ travels along the geodesic with this construction. To this end, denote $\beta_{ki}^l := \sphericalangle(x_{ki}^l, \bar{x}_{ki})$ as the spherical angle between $x_{ki}^l$ and $\bar{x}_{ki}$. Let $\Delta\beta_{ik}^l := \beta_{ki}^{l+1} - \beta_{ki}^l$, i.e., the angle shift of $x_{ki}^l$ in the $l$-th layer of the $ki$-th block. Using simple trigonometry we can compute:

$$\Delta\beta_{ik}^l = 2\arcsin\left(\frac{\alpha_{ki}^l m}{2\sqrt{d+m^2/4}}\right) \geq \frac{m}{4\sqrt{d}}\alpha_{ki}^l,$$

where the inequality holds for $\alpha_{ki}^l$ small enough.

Therefore, it suffices to choose $L$ and $L_1$ large enough and set $\alpha_{ki}^l = \frac{4\sqrt{d}\beta_{ki}^l}{L_1 m}$ if $(x_{ki}^l)^T h_{ki} \leq d(1-cm)$ and $0$ otherwise. In this way, the total regularization cost of the layers in the first $N$ blocks can be upper bounded as

$$\frac{\lambda}{2}\sum_{k,i,l}^{K,n,L_1}\|W_{ki}^l\|_F^2 \leq \frac{32d\pi^2\lambda N}{L_1 m^2}.$$

We see that this cost goes to $0$ as $L_1$ goes to infinity.

After $N$ blocks, all the samples now lie within the $c$-multiple of margin $((x_{ki}^{L_1})^T h_{ki} \geq d(1-cm))$ of their respective optimal $h_{ki}$ features. The goal of each of the $\bar{K}$ blocks is to move the corresponding samples in the $j$-th group all together ever closer to these final vectors. Since this time the construction will be equivalent for all the samples within one group, we will refer to these samples simply as a single $j$-th sample in the $l$-th layer of the respective block, using the notation $x_j^l$. We define all layers in the $j$-th block as follows:

$$W_j^l = \alpha_j^l \frac{1+cm\bar{h}_j}{\|1+cm\bar{h}_j\|}\frac{\bar{h}_j^T}{\sqrt{d}},$$

$$b_j^l = -(1-2cm)\frac{\alpha_j^l\sqrt{d}}{\|1+cm\bar{h}_j\|}\mathbf{1},$$

where again $\alpha_j^l$ is an optimizable parameter. By similar computations as above, the above construction makes sure that $\sigma(W_j^l x_j^l + b_j^l) = \frac{\alpha_j^l}{\|1+cm\bar{h}_j\|}((\bar{h}_j^T x_j^l - d + 2cmd)\mathbf{1} + cm\bar{h}_j^T x_j^l \bar{h}_j)$ while $\sigma(W_j^l x_i^l + b_j^l) = 0$ for $i \neq j$. After subtracting the mean in the layer norm we are adding $\frac{\alpha_j^l cm\bar{h}_j^T x_j^l}{\|1+m\bar{h}_j\|}\bar{h}_j$, which is at least a $\frac{\alpha_j^l cm}{2\sqrt{d}}$ multiple of $\bar{h}_j$. Denote $\beta_j^l = \sphericalangle(x_j^l, \bar{h}_j)$ and $\Delta\beta_j^l = \beta_j^{l+1} - \beta_j^l$.

Using trigonometry again, we get:

$$\Delta\beta_j^l \geq \arctan\left(\frac{\alpha_j^l cm\sin(\beta_j^l)}{2\sqrt{d}(1+\alpha_j^l cm\cos(\beta_j^l)/(2\sqrt{d}))}\right) \geq \frac{\alpha_j^l cm\sin(\beta_j^l)}{4\sqrt{d}(1+\alpha_j^l cm\cos(\beta_j^l)/(2\sqrt{d}))} \geq \frac{\alpha_j^l cm\beta_j^l}{16\sqrt{d}},$$

where all inequalities hold from basic properties of trigonometric functions for small-enough angles. Thus, the angular shift is lower bounded as follows: $\Delta\beta_r^l \geq \frac{\alpha_j^l cm\beta_j^l}{16\sqrt{d}}$. If we choose $\alpha_j = \alpha_j^l$ constant across layers, we get $\beta_j^{L_2} \leq \left(1 - \frac{\alpha_j cm}{16\sqrt{d}}\right)^{L_2}\beta_j^0$.

We will choose $\alpha_j = \frac{16\sqrt{d}\log(L_2)}{cmL_2}$. Then, the total regularization of the layers in the penultimate blocks is upper bounded as follows:

$$\frac{\lambda}{2}\sum_{l,j=1}^{L_2,\bar{K}}\|W_j^l\|_F^2 \leq \frac{2^7 Nd\lambda\log(L_2)^2}{m^2 L_2}.$$

This goes to zero linearly up to poly-log factors as $L_2$ goes to infinity. Finally, we have that the final positions $x_{ki}^{L_2}$ of the samples converge fast to their optimal counterparts $h_{ki}$ with $L$. To see this, plugging our choice of $\alpha_j$ into $\beta_j^{L_2} \leq \left(1 - \frac{\alpha_j cm}{16\sqrt{d}}\right)^{L_2}\beta_j^0$ we get $\beta_j^{L_2} \leq \left(1 - \frac{\log(L_2)}{L_2}\right)^{L_2}\beta_j^0 \leq \frac{2\beta_r^0}{L_2}$, so the samples converge linearly to their optimal positions as $L_1, L_2 \to \infty$. From the continuity of

the fit part of the loss, we see that the total loss of this construction indeed converges to $\mathcal{L}^*_{\text{GUFM}}$ of the corresponding GUFM problem. Therefore, our upper bound on the loss of globally optimal solutions converges to $\mathcal{L}^*_{\text{GUFM}}$ and evoking Lemma 4.1 we know that a $(W_L, X_L)$ optimal for (7) is nearly optimal for (8) and thus exhibits the required convergence.

Next, we continue with the proof for $L$-Tx1. Notice that, if we can ensure after the end of the first block that all the *different* contexts have different representations and that two representations of different contexts don't lie on a line with some of the final positions $h_{ki}$, then by setting all the weights in attention layers of the subsequent blocks to 0, the rest of the transformer becomes a ResNet with LayerNorm and we can apply an identical construction as in the $L$-RN1 part to conclude. The only caveat (except making sure that the margin is positive) is that, since the total regularization loss of the construction for $L$-RN1 goes to 0 with $L$ going to infinity, we must make sure that the same is true for the first block. However, as we will see, this will make the margin $m$ a function of $L$ that slowly goes to 0. To compensate for this, we will need to set the layers in the subsequent blocks accordingly bigger, and we will make sure that the margin goes to 0 slowly enough so that this adjustment will not qualitatively change the results. Another issue we have to deal with is that if the norm of the $W_{QK}$ matrix has to go to 0, the attention weights must necessarily converge to uniform. Thus, our construction must withstand this burden.

We will start with the construction of the embedding matrices. The embedding matrix $W_e \in \mathbb{R}^{d \times d_0}$ will just lift the dimension to the inner-dimension of the transformer $d_l \geq 2V + 2$, i.e., the $v$-th column of $W_e$ is $e_v$ in $d_l$-dimensional space. Then, the $(C - i)$-th column of $W_p \in \mathbb{R}^{d \times C}$ will be $a \cdot e_{2V+1} + b \cdot e_{2V+2}$, where $a, b > 0$ are the unique solutions of the following two equations: $a + b = -1$ and $a^2 + b^2 = 2^{2(i+1)} - 1$. Thus, after the embedding layer, the sum of the entries of the entire embedding is 0 and after the first normalization layer, the $j$-th token at the $(C - i)$-th position will have $\sqrt{d}2^{-(i+1)}$ on its $j$-th entry and the only other non-zero entries will be at positions $2V + 1$ and $2V + 2$.

Let us construct the first block. Here, all MLP layers will be set to 0 so that they have zero effect. Moreover, due to the constraints discussed above, attention matrices $W_K, W_Q$ or $W_{QK}$ will also be set to zero. Finally, the value and output matrices will be set as $W_V = W_O = \sqrt{\gamma(L)}A$ or $W_{VO} = \gamma(L)A$, where $A$ shifts all entries from the range $1, \ldots, V$ to the range $V + 1, \ldots, 2V$, and $\gamma(L)$ is a decreasing function converging to 0 at infinity that will be defined later. This ensures that the representations before and after attention are summed in the residual connection on different positions, which will be technically convenient later. Since the attention matrices are identically zero, the attention weights corresponding to the $c$-th token will just be uniform $1/c$ for all the tokens up to this one. Therefore, the representation of the $c$-th token after the attention layer and before the residual connection is the $\gamma(L)$-multiple of the average of all the representations of the previous tokens and itself from an input to the first block shifted by $V$ positions.

We now show that two different contexts must necessarily have different representations, which gives that the margin after block 1 is non-zero. If we compare two samples (contexts) with different context lengths, then they will necessarily have different numbers of distinguishable summands (i.e. various negative powers of 2, divided by the sample's context length) present in the entries between $(V + 1)$-th and $2V$-th. Since there is a different number of summands, there must exist at least one entry where the number of summands disagree, and the numbers in this entry must have different numbers of ones in their binary representation, which guarantees that samples with different context lengths must have different representations. Furthermore, two samples with the same context length but different contexts will be divided by the same averaging number, but then they can be distinguished since the map from contexts to representations (without dividing by the context length) is injective due to the uniqueness of the binary representation of the summands.

Therefore, all non-identical contexts have different representations and, in addition, the previous argument also shows that every pair of representations of two different contexts is linearly independent. This remains true after the residual connection. If we choose $\gamma(L)$ small enough for all $L$, then the original encodings of the current token will not mix up with the much smaller summands from the attention layer. The relative size of all the summands stays the same also after normalization and the MLP block has no effect, so all different contexts have different representations after the first layer. The only issue we could face is that the representations end up coinciding with one of the $h_{ki}$'s. To avoid this, $W_O$ or $W_{VO}$ can implement a tiny rotation. Since the number of tiny rotations is uncountably infinite, there is at least one for which there is no intersection. Let $\sqrt{d}\tilde{m}$

be the minimal distance between representations of any two samples after the attention mixing, before the multiplication by value and output matrices and before the residual connection. Note that $\tilde{m}$ is positive and independent of $X, Y, L$, because the different contexts are all pairwise linearly independent. Then, after the multiplication by the value and output matrices, such distance will be $\gamma(L)\sqrt{d}\tilde{m}$. For small enough $\gamma(L)$, the worst-case addition in the residual connection corresponds to the case in which the two samples with the same latest token also realize the margin minimum. However, if $\gamma(L) \leq 0.1$, then the difference of the samples after the residual connection and after the normalization is at least equal to the distance between the representations on positions $V + 1$ to $2V$, which is at least $\gamma(L)\sqrt{d}\tilde{m}$. Thus, this is the minimum pairwise distance of the data after the first attention block.

Next, we can apply the construction for $L$-RN1 if we set all the remaining attention layers to zeros, since then the remainder of the network will be functionally equivalent to $L$-RN1. The only remaining issue is that the margin after the first layer is a function $\gamma(L)$ of the total number of layers. To choose a good scaling of $\gamma(L)$, we need to consider the elements of the construction for $L$-RN1 that depend on the margin, which is the sum of the Frobenius norms of the layers in the first $N + \bar{K}$ blocks. This is upper-bounded by $\frac{32\pi^2\lambda Nd}{L_1 m^2} + \frac{128Nd\lambda \log(L_2)^2}{m^2 L_2}$. Therefore, if we choose $\gamma(L) = \Theta(L_1^{1/4}) = \Theta(L_2^{1/4})$, then both the sum of Frobenius norms of the layers in first $N$ layer blocks, as well as the Frobenius norms of $W_V, W_O$ or $W_{VO}$ in the first block of the transformer will go to 0 as $L_1, L_2 \to \infty$. This concludes the proof. $\square$

**Corollary 4.5.** *Let the architecture be L-RN1 or L-Tx1 for $x \in \{1, 2\}$. Assume the training data $(X, Y)$ satisfies Assumption 4.4 and all the assumptions of Theorem 4.3. Using CE or MSE loss, all global optima of the optimization problem (7) exhibit approximate neural collapse which gets tighter as $L$ increases.*

*Proof.* This is a straightforward combination of Lemma 4.2 and Theorem 4.3 once we use that identical contexts for transformers are only labeled by one class, which allows to directly apply the lemma. $\square$

**Theorem 4.6.** *Let the architecture be L-RN2 or L-Tx2 for $x \in \{1, 2\}$. Assume the inner dimension of the L-Tx1 is at least $2V + 2$ and the inner dimension of L-RN1 is at least 4. Consider the optimization problem*

$$\min_{\theta} \mathcal{L}(f_\theta(X), Y) + \frac{\lambda_L}{2}\|W_L\|_F^2 + \frac{\lambda(L)}{2}\|\bar{\theta}\|^2, \tag{11}$$

*where $\lambda_L$ is a regularization on the weight matrix of the last layer that does not depend on $L$ and $\lambda(L)$ is a depth-dependent regularization s.t. $\lambda(L) = o(\log(L)^{-1})$. Consider the corresponding GUFM with regularization $\lambda_L$. If $\mathcal{L}_{GUFM}^* > 0$, then*

$$\limsup_{L \to \infty} \text{distmax}(\tilde{\mathcal{M}}_L \backslash \mathcal{M}_0^{GUFM}, \mathcal{M}_0^{GUFM}) = 0. \tag{12}$$

*Proof.* The proof follows that of Theorem 4.3. The only difference is that the construction of the weight matrices changes so that $W_{ki}^{l,1}$ and $b_{ki}^{l,1}$ have $\sqrt{\alpha_{ki}^l}$ in place of $\alpha_{ki}^l$. The second layers' weight matrices $W_{ki}^{l,2}$ are defined as $\sqrt{\alpha_{ki}^l}$-multiples of the projection matrix on the span of the output of the first sub-layer on sample $x_{ki}^l$, so that the total mapping will be identical to the single-layer construction. Using analogous computations as above, we get:

$$\frac{\lambda(L)}{2}\sum_{k,i,l}^{K,n,L_1}\left\|W_{ki}^{l,1}\right\|_F^2 + \left\|W_{ki}^{l,2}\right\|_F^2 \leq \frac{16\sqrt{d}\pi\lambda(L)N}{m},$$

and for the second part of the blocks we get:

$$\frac{\lambda}{2}\sum_{l,j=1}^{L_2,\bar{K}}\left\|W_j^{l,1}\right\|_F^2 + \left\|W_j^{l,2}\right\|_F^2 \leq \frac{16N\sqrt{d}\log(L_2)\lambda(L)}{m}.$$

In order for the sum of these two components to go to zero, we need $\lambda(L) = o(\log(L_2)^{-1})$ and we can choose $L_1 = \Theta(L_2)$. The rest of the proof is identical to that of Theorem 4.3. $\square$

**Corollary 4.7.** *Let the architecture be L-RN2 or L-Tx2 for $x \in \{1, 2\}$. Assume the training data $(X, Y)$ satisfies Assumption 4.4 and the assumptions of Theorem 4.6. Using CE or MSE loss, all global optima of the optimization problem (11) exhibit approximate neural collapse which gets tighter as $L$ increases.*

*Proof.* This is a straightforward combination of Lemma 4.2 and Theorem 4.6 once we use that identical contexts for transformers are only labeled by one class, which allows to directly apply the lemma. $\square$

# B    Alternative architectures

## B.1    Vision transformers

For vision transformers, the data is tensor-like $X_0 \in \mathbb{R}^{N \times d_0 \times C}$, where $C$ now denotes the number of patches and $d_0$ is the dimension of the patch. However, the labels remain two-dimensional $Y \in \mathbb{R}^{N \times K}$. What is considered as a sample depends on how labels are produced in the transformer. The simplest option (w.r.t. the rest of our paper) is to generate the prediction on the last patch of the sequence, keeping the causal mask. This will, however, change the definition of "samples" and the NC metrics, since we only need to focus on the last patch. Therefore, samples will only be considered as the last patch, and the NC metrics will only be defined over the representations of the last patches. Similarly, the equivalent DUFM will also correspond to the last patches.

Theorem 4.3 and 4.6 and, thus, also Corollary 4.5 and 4.7 hold for vision transformers too, as long as we do the following changes to the proof of Theorem 4.3 (the other statements are adjustable trivially once this is established).

**Necessary adjustments to the proof.**    Together with the uniqueness of the labeling function, we will also assume that the samples are taken from a continuous distribution (which is reasonable in the vision domain). This guarantees that the feature representations of the final patches are unique also after the first transformer block, as the event that averages over patches of two different samples coincide has zero probability. The rest of the proof is similar to that of Theorem 4.3, but the subsequent MLP layers only focus on the movement of the last patches' representations and the movement of the other patches is irrelevant.

## B.2    Pre-LN ResNets and transformers

Unlike the post-LN ResNets (Definition 3.1) and transformers (Definition 3.2), the pre-LN architectures apply the LayerNorm directly before the attention and/or linear layers, but only *within* the residual connection, leaving the main residual stream untouched. While this potentially makes the features at initialization grow linearly with depth, it makes for more stable gradients thanks to the direct residual path, avoiding LayerNorms that can serve as error propagation channels. This significantly simplifies the training dynamics and therefore the pre-LN transformers are currently being predominantly used. For this reason, we fully define the pre-LN architectures here and then discuss in sufficient amount of detail how to adjust the proof for this setting, since the results are qualitatively the same.

**Definition B.1.** *An L-block pre-LN ResNet with LayerNorm and one linear layer per block (later referred to as pre-L-RN1) is defined as*

$$f_\theta = \lin_L \circ \mathrm{LN} \circ (\mathrm{id} + \sigma \circ \lin_{L-1} \circ \mathrm{LN}) \circ (\mathrm{id} + \sigma \circ \lin_{L-2} \circ \mathrm{LN}) \circ \cdots \circ (\mathrm{id} + \sigma \circ \lin_1 \circ \mathrm{LN}) \circ \mathrm{LN} \circ \lin_0, \tag{15}$$

*where $\lin_l(x) = W_l x + b_l$ for all $l \in \{0, \dots, L\}$ and $\theta$ is the collection of all learnable parameters. We denote as $X_1 = \mathrm{LN}(W_0 X_0 + b_0)$, $X_{l+1} = X_l + \sigma(W_l \mathrm{LN}(X_l) + b_l)$ ($l \in \{1, \dots, L-1\}$), $f_\theta(X_0) = X_{L+1} := W_L \mathrm{LN}(X_L)$ the intermediate representations of the training data stored in a matrix form. We assume that all intermediate representations $X_l$ ($l \in \{1, \dots, L\}$) are of dimension $d$. Analogously, L-RN2 denotes a ResNet with two linear layers per block defined as*

$$f_\theta = \lin_L \circ \mathrm{LN} \circ (\mathrm{id} + \lin_{L-1,2} \circ \sigma \circ \lin_{L-1,1} \circ \mathrm{LN}) \circ \cdots \circ (\mathrm{id} + \lin_{1,2} \circ \sigma \circ \lin_{1,1} \circ \mathrm{LN}) \circ \mathrm{LN} \circ \lin_0, \tag{16}$$

*with $X_1 = \mathrm{LN}(W_0 X_0 + b_0)$, $X_{l+1} = X_l + W_{l,2}\sigma(W_{l,1}\mathrm{LN}(X_l) + b_{l,1}) + b_{l,2}$ ($l \in \{1, \dots, L-1\}$) and $f_\theta(X_0) = X_{L+1} := W_L \mathrm{LN}(X_L)$.*

**Definition B.2.** *An L-block pre-LN transformer with one or two linear layers in the attention sub-block and one or two layers in the MLP sub-block (later referred to as pre-L-T11, pre-L-T12, pre-L-T21, pre-L-T22 based on the number of linear layers in attention and MLP sub-blocks, respectively) is defined as*

$$f_\theta(Z) = \lin_{L+1} \circ \mathrm{LN}_{L+1} \circ \mathrm{B}_L \circ \cdots \circ \mathrm{B}_1 \circ \mathrm{LN}_0 \circ \mathrm{Embed}(Z). \tag{17}$$

*Here,* $\lin_{L+1}(Z) = W_{L+1}Z + b_{L+1}$ *is the last layer ($b_{L+1}$ is a matrix with the same number of columns as $Z$ that are all identical);* $\mathrm{Embed}(Z) = W_e Z + W_p$ *is the embedding layer with $W_e$ being the token embedding and $W_p$ (having the same shape as $W_e Z$) the positional embedding; and the l-th block is given by*

$$\mathrm{B}_l = (\mathrm{id} + \mathrm{MLP}_l \circ \mathrm{LN}_{l,2}) \circ (\mathrm{id} + \mathrm{ATTN}_l \circ \mathrm{LN}_{l,1}). \tag{18}$$

*Such block consists of the normalization layers* $\mathrm{LN}_{l,1}, \mathrm{LN}_{l,2}$*, the MLP*

$$\mathrm{MLP}_l(Z) = \sigma(W_l Z + b_l), \ or \ \mathrm{MLP}_l(Z) = W_{l,2}\sigma(W_{l,1}Z + b_{l,1}) + b_{l,2}, \tag{19}$$

*respectively for the architecture pre-L-Tx1 and pre-L-Tx2, and the single-head attention*

$$\mathrm{ATTN}_l(Z) = W_{VO}ZA_l(Z), \ A_l(Z) = \mathrm{softmax}(M + Z^T W_{QK} Z/\sqrt{d}),$$
$$or \ \mathrm{ATTN}_l(Z) = W_O W_V Z A_l(Z), \ A_l(Z) = \mathrm{softmax}(M + Z^T W_K^T W_Q Z/\sqrt{d}), \tag{20}$$

*respectively for the architecture pre-L-T1x and pre-L-T2x. The matrix $M$ is the masking matrix whose entries are $-\infty$ on the lower triangle and $0$ on the upper triangle and the diagonal.*

We note that the first LayerNorm right after the embedding layer, which might not be used in practice often, is introduced for technical convenience but does not change the results qualitatively. Theorem 4.3 and 4.6 and, thus, also Corollary 4.5 and 4.7 hold for pre-LN architectures too, as long as we do the following changes to the proof of Theorem 4.3 (the other statements are adjustable trivially once this is established).

**Necessary adaptations to the proof.** This architecture has the disadvantage that it does not immediately absorb deviations from the zero-sum sphere and therefore, technically, the single linear layer architectures can only add non-negative changes to the residual stream. However, we argue that an *almost identical* construction to the one in proof of Theorem 4.3 works here as well. Note that the construction from this proof, see (14):

$$W_{ki}^l = \alpha_{ki}^l \frac{1 + \frac{m}{2}d_{ki}^l}{\sqrt{d + m^2/4}} \frac{(x_{ki}^l)^T}{\sqrt{d}},$$

$$b_{ki}^l = -\left(1 - \frac{m}{2}\right)\frac{\alpha_{ki}^l \sqrt{d}}{\sqrt{d + m^2/4}}\mathbf{1},$$

will result in a shift in $x_{ki}$ that can be written as $\frac{\alpha_{ki}^l m\sqrt{d}}{2\sqrt{d+m^2/4}}(\mathbf{1} + d_{ki}^l)$ and the $\mathbf{1}$ will not get absorbed in the residual stream, but *is orthogonal* to the zero-sum component of the movement of $x_{ki}$ *and it will* get absorbed in the next LayerNorm within the next residual stream. This allows us to copy the entire first part of the post-LN proof by mimicking the trajectories of the unit ball, while adding the constant amount of $\frac{\alpha_{ki}^l m\sqrt{d}}{2\sqrt{d+m^2/4}}$—multiple of all-ones vector in each round. Therefore, after the first $N$ blocks, the projections of all the samples on the zero-sum hyperplane are identical to those in the post-LN proof. Each sample, however, has a different component in the direction of the all-one vector. This will, however, be absorbed by the last LayerNorm. Moreover, by triangle inequality, the margin of the trajectories in this extended space is at least as big as the margin of the trajectories on the zero-mean ball. The construction for the next $\tilde{K}$ blocks works by the same reasoning as well. Thus, after these layers, the projections of the samples on the zero-sum ball are identical to the post-LN proof and the last LayerNorm will absorb the component along the all-ones vector.

As for the transformers, although after the first block the samples are not centered and do not all have norm $\sqrt{d}$, after applying the LayerNorm in the first subsequent MLP block, they will all be distinct (except the ones with identical contexts). Therefore, we define the same trajectories as in the ResNet construction with the centered and normalized features, but we will perform an equivalent movement on the zero-mean ball with the radius equal to the norm of the projection of the particular

sample onto the zero-mean hyperplane, while ignoring the all-ones component completely. As a result, each sample moves on its own cylinder, a projection to the zero-mean hyperplane following the trajectories on the normalized zero-mean ball, while moving arbitrarily along the all-ones direction. As before, a triangle inequality guarantees that the margin defined on the zero-mean normalized ball is not violated in the wider space during this process. The only caveat is that, if the norm of the ball along which a sample is traveling is larger than that of the $\sqrt{d}$-normed ball, we need to upscale $\alpha_{ki}^l$ by that ratio. Note that the size of the vector after the first block is upper bounded independently of the number of layers, therefore such an upscaling will only multiply the cost of weight matrices by a constant. The rest of the argument follows that of the adaptation for pre-LN ResNets.

## C  Two linear layers per block with non-vanishing or uniform weight decay

Here, we intuitively describe why the NC metrics in general *do not* approach the perfect NC in architectures with two linear layers per residual block as the depth goes to infinity, *if the regularization is non-vanishing or vanishes uniformly across all layers*. The key is the simple inequality $\|AB\|_F \leq \|A\|_F \|B\|_F$. We can interpret these matrices as features, weight matrices and the change on the features added to the residual. In particular, in a ResNet with a single linear layer per block we have $\|\Delta X_l\|_F \leq \|W_l\|_F \|X_l\|_F$ ($\Delta X_l$ is the outcome of the residual branch added back to the residual stream) and, importantly, this inequality can be made equality in some cases. Even if the inequality does not hold as an equality, we still have that, for fixed $W_l, X_l$, if $\Delta X_l = \sigma(W_l X_l) \neq 0$, then due to homogeneity $c\Delta X_l = (cW_l)X_l$. This makes the total change $W_l$ makes to $X_l$ scale linearly with $c$, but its cost is quadratic. Therefore, if the directional derivative of the loss w.r.t. $\Delta X_l$ at layer $l$ is strictly positive, then there exists $c > 0$ for which $cW_l$ *will* make an improvement against $W_l = 0$. However, if two linear layers are involved, we have

$$\|\Delta X_l\|_F \leq \|W_{l,2}\sigma(W_{l,1}X_l)\|_F \leq \sqrt{N/4}\left(\|W_{l,1}\|_F^2 + \|W_{l,2}\|_F^2\right).$$

Therefore, any change to the features will scale *linearly with the regularization cost* of the matrices that were responsible for this change. In this case, the opposite to the previous statement holds: if the directional derivative of the loss w.r.t. $\Delta X_l$ is *small enough,* then for small $c$, the $c$-scaling of weight matrices will necessarily *worsen* the loss compared to doing nothing.

As we have seen in the proof of Lemma 4.2 for the MSE loss (but this also holds for the CE loss), an $\mathcal{O}(\epsilon)$-sized perturbation around the global optimum of neural collapse causes only an $\mathcal{O}(\epsilon^2)$ increase in the loss. Furthermore, the derivative is zero at NC and locally Lipschitz around that point, which implies that the size of the derivative is $\mathcal{O}(\epsilon)$. For any input dataset $X$ that is not yet collapsed, if the points $W_L, X_L$ in the set of global optima $\tilde{\mathcal{M}}_{L,2}$ *did* approach NC in the limit, we could, by contradiction, take an optimum that is $\epsilon$-close to NC (for a sufficiently small $\epsilon$) and zero-out all the last layers that were responsible for moving the samples by a total amount of $\Theta(\epsilon)$ shift (this would need care in a rigorous proof because of the possible discontinuity of the layer-to-feature mapping). The change in the fit part of the loss would be $\mathcal{O}(\epsilon^2)$, but thanks to the above inequality, the total regularization cost saved by this would be $\Omega(\epsilon)$, so the loss would improve and we would arrive at a contradiction.

The above argument holds for constant regularization $\lambda$. However, even if the regularization was vanishing, but it was the same for $W_L$ and for the rest of the network, the NC would still not be approached. To see this for MSE loss, consider a perturbed perfect scenario where the input data is $X = I_K \otimes \mathbf{1}_n^T + E$ and $E$ is a perturbation matrix of size $\Theta(\epsilon)$. $X$ is already $\epsilon$-close to NC. To move $X$ $\Theta(\epsilon)$ closer to NC, we need $\Theta(\lambda_L \epsilon)$ cost in terms of the weight matrices. Let us now compute the improvement in the corresponding GUFM objective that results from doing so. The DUFM objective with MSE is $\frac{1}{2N}\|WX_L - Y\|_F^2 + \frac{\lambda_L}{2}\|W\|_F^2$. If we simplify the problem to just fitting a single row of $W$ (the optimization problem is separable, so this is w.l.o.g.), we have a simple ridge regression solution for $w$. In particular

$$w^* = (nI_K + \lambda_L I_K + \mathcal{O}(\epsilon))^{-1}(I_K \otimes \mathbf{1}_n^T + \mathcal{O}(\epsilon))y.$$

Therefore, the distance from the unperturbed fit $(nI_K + \lambda_L I_K)^{-1}(I_K \otimes \mathbf{1}_n^T)y$ is itself $\mathcal{O}(\epsilon)$ and plugging this in the loss, we see that the change in the loss function is $\mathcal{O}(\lambda_L \epsilon^2)$ which, for sufficiently small $\epsilon$, is less than the price in terms of weight regularization.

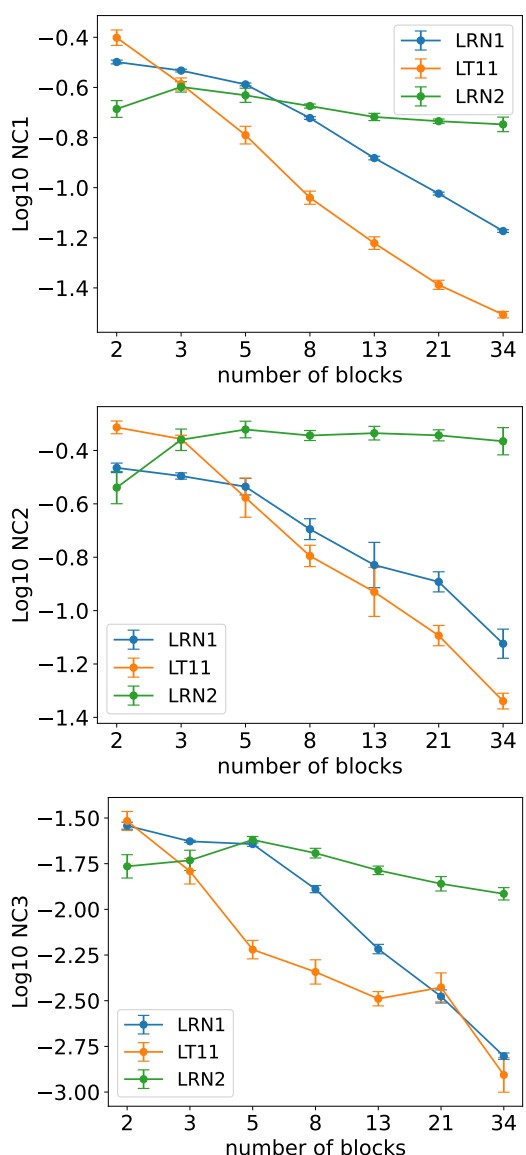

Figure 2: MNIST training. $\log_{10}$ of NC1, NC2 and NC3 metrics respectively in the upper, middle and bottom row, as a function of the number of blocks $L$. The architectures are $L$-RN1 with $\lambda = 0.005$, $L$-T11 with $\lambda = 0.005$, and $L$-RN2 with $\lambda = 0.0025$.

## D   Additional experimental results

In Figure 2, we provide additional experimental results which complement Figure 1 with MNIST training of both ResNets and transformers with one linear layer per MLP block. The results and the message are consistent with those of Section 5. Furthermore, we consider a ResNet with two linear layers per block trained on MNIST, but with constant weight decay of $0.0025$. As we can see, the NC metrics are almost constant across multiple depths, which is consistent with our claim that NC is not approached in this regime.

Additionally, in Figure 3 we show the sum of Frobenius norms of all the weight matrices of ResNets trained on MNIST. We can see that as the number of layers increases, the total norm of the weights decreases. We note that this is the same phenomenon leading to neural collapse in our theory. Similar results hold for the other training scenarios as well.

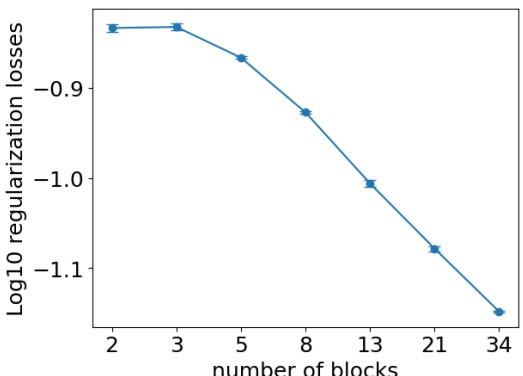

Figure 3: $\log_{10}$ of total regularization loss of ResNet trained on MNIST as a function of the depth of the network.

