# OpenReview forum: "Neural Collapse is Globally Optimal in Deep Regularized ResNets and Transformers"
_NeurIPS.cc/2025/Conference — NeurIPS 2025 poster_

### Official Review · Reviewer_agMG · 2025-06-26

**Clarity:** 3
**Significance:** 4
**Originality:** 4
**Rating:** 4
**Confidence:** 5

**Summary:**

This paper investigates neural collapse (NC) in deep regularized ResNets and transformers with LayerNorm, trained end-to-end with cross-entropy or mean squared error loss. It proves that global optima of these architectures exhibit approximate NC as depth increases, extending prior work limited to multi-layer perceptrons. By reducing the training objective to a generalized unconstrained features model (GUFM), it shows that NC is asymptotically optimal, with the distance to perfect collapse bounded by depth. The results hold under minimal data assumptions and are supported by experiments on vision datasets, demonstrating increased collapse with depth. This work provides a theoretical justification for using UFMs to analyze modern architectures.

**Questions:**

No

**Ethical Concerns:**

["NO or VERY MINOR ethics concerns only"]

**Limitations:**

Yes

**Quality:**

3

**Strengths And Weaknesses:**

**Strengths**: The paper offers a groundbreaking theoretical analysis of neural collapse in modern architectures like ResNets and transformers, significantly advancing beyond prior MLP-focused studies. Its rigorous reduction of end-to-end training to a GUFM elegantly connects complex architectures to simpler models, providing novel insights into NC optimality. The innovative proof techniques, and the clear justification for UFM applicability make this a contribution to understanding feature representations in deep learning.

**Weakness**:
When I saw that the global optima of deep models with CE loss could be formulated, I was thrilled. I had once tried to characterize the gradient descent dynamics of MLPs, where NC1, NC2, and NC3 gradually decrease with increasing layers and training iterations, but gave up due to its complexity. However, I was somewhat disappointed to see this paper’s results rely on constructive proofs. While not a flaw, I believe this limits the work’s practical utility. Due to the well-known implicit bias, when multiple optimal solutions exist, optimization algorithms tend to favor certain solutions with better generalization. The authors should add experiments to verify if the training dynamics of cross-entropy loss in deep models align with the constructed solutions (which I doubt) and explore, through experiments, how training dynamics select intermediate layer parameters to achieve NC if they deviate.

Given that this addition isn’t central to the paper’s main contribution and rebuttal time is limited, I believe the authors should, at minimum,
make some experiments to verify if the constructive solution will be selected by optimization algorithm  and discuss these shortcomings thoroughly in the conclusion and highlight corresponding challenges to inspire future work.

---

> ### Author Rebuttal · Authors · 2025-07-30
>
> We thank the reviewer for the in-depth review and the overall positive evaluation. We address the reviewer’s concerns and questions below.
>
> ---
>
> **Verify if the constructive solution will be selected by SGD and discuss these shortcomings.**
>
> We do not necessarily expect the solutions constructed in our proofs to be achievable by gradient descent. The construction moves one sample per layer, and for finite depths it serves as an upper bound to the loss exhibited by global optima. In contrast, we expect both the global optima, as well as the architectures trained via gradient descent, to move multiple samples simultaneously. However, as experimentally measured in Figures 1 and 2, we *do* expect the solutions found by gradient descent to exhibit neural collapse ever more exactly with increasing depth. In fact, we expect an asymptotic rate of convergence to neural collapse which is similar to that exhibited by our constructive solution. To support this claim, we have computed the slopes of the NC curves in Figures 1 and 2 for networks with one linear layer per residual block (computed starting from depth 5, since the shallower depths are too weak to follow the trend), and we have found that such slopes average at around $-0.335$. This is comparable to our theory, which ranges between $-1/2$ and $-1/4$, depending on the setting. We will clarify this in the revision.
>
> ---
>
> **Explore, through experiments, how training dynamics select intermediate layer parameters to achieve NC.**
>
> This is an interesting question. An extensive answer is an open problem and a great future direction (we will mention this in the conclusion). We can still give partial observations, though. First, due to the limited number of layers ($\le 34$) compared to the number of samples ($50000$), it is clear that the layers are moving all or at least a high proportion of samples in each layer. Second, although the solution we construct is likely not achieved by gradient descent, the main takeaway of our proof (layers getting smaller, splitting the total work, so that they decrease their total norm) remains true in practice. To support this claim, we have computed the sum of all layers’ squared Frobenius norms multiplied by a constant regularization parameter for ResNet trained on MNIST (corresponding to the experiment of Figure 2 of the Appendix), as a function of the depth of the network.
>
> Number of layers: [2, 3, 5, 8, 13, 21, 34]
>
> Means: [0.1467, 0.1471, 0.1360, 0.1182, 0.0986, 0.0835, 0.0711]
>
> Standard deviations: [0.0015, 0.0014, 0.0004, 0.0005, 0.0010, 0.0006, 0.0002]
>
> This clearly shows that, as the number of layers increases, the total norm of the weights decreases. We note that this is the same phenomenon leading to neural collapse in our theory. Similar results hold for the other training scenarios as well. We will add a figure in the revision.

---

### Official Review · Reviewer_EMH7 · 2025-07-01

**Clarity:** 2
**Significance:** 3
**Originality:** 3
**Rating:** 5
**Confidence:** 1

**Summary:**

This article provides a theoretical analysis that proves that under weight regularization and standard loss functions (MSE and CE),  in Transformer and Resnet architectures, the optimal solution is approximately equivalent to Neural Collapse. And as depth increases, this approximation gets tighter.

**Questions:**

As a practitioner, I find it hard to quantify the importance of this theoretical work. Could you expand on why the modeling of Neural Collapse is important and what takeaways a practitioner should take? Do we always want to have Neural Collapse when training neural networks? Does this work tell us that with more depth, we can expect better generalization on these main architectures?

**Ethical Concerns:**

["NO or VERY MINOR ethics concerns only"]

**Final Justification:**

I maintain the positive review as I think this paper builds on the understanding of the interesting phenomena of neural collapse. However due to the theoretical nature of the paper that is not in my area, I found it hard to follow the theory, as such my confidence score is low.

**Limitations:**

yes

**Quality:**

3

**Strengths And Weaknesses:**

Strengths:
+ The article is well written
+ connects the impact of depth and neural collapse on 2 well-established architectures
+ The empirical results follow the theory

Weaknesses:
+ As a practitioner, from reading this paper alone is hard to obtain the main takeaways from this theoretical work, for example, on what neural collapse is and why it is important.

---

> ### Author Rebuttal · Authors · 2025-07-30
>
> We thank the reviewer for the positive evaluation of our paper. We are happy to address the reviewer's practical concerns about the applicability of neural collapse and we will extend our related work section in the revised version to include a more elaborate explanation concerning this point.
>
> ---
>
> **Why is it important to model the neural collapse and what are the practical implications?**
>
> Neural collapse is a high-impact phenomenon, with the seminal paper [1] that introduced it reaching over 700 citations, the majority of which are practical applications of neural collapse. These include (but they are not limited to) transfer learning, OOD detection, compression and performance improvement. Neural collapse also helped theoreticians establish compelling results in generalization, OOD detection, feature learning, robustness and other areas. Please refer to lines 25-31 of our introduction for citations.
>
> The vast applicability of neural collapse makes it highly desirable to characterize. This has prompted a line of theoretical work aimed at providing a precise understanding of the architectures/conditions under which we can expect to achieve neural collapse (and, conversely, for which architectures/conditions, neural collapse is unlikely to appear).
>
> ---
>
> **Do we always want to have the neural collapse when training neural networks?**
>
> This is a great question. While works that apply neural collapse need it and some theoretical works suggest it can have a positive impact on the performance of neural networks [1, 2, 3, 4, 5], neural collapse is not always helpful. In particular, NC is only fully achieved if the training loss is very close to zero. Therefore, in scenarios where overfitting could be an issue and we would need to apply early stopping strategies or deliberately short training, we in fact want to avoid the emergence of neural collapse. Understanding under which conditions neural collapse is desirable is important and theoretical analyses can help us answer this question.
>
> ---
>
> **Does this work tell us that with more depth, we can expect better generalization on these main architectures?**
>
> While some works claim a connection between neural collapse and better generalization [1, 2, 3, 5], a formal, rigorous link between collapse and generalization has not yet been established. Therefore, our work does not conclusively tell us whether better generalization can be expected in deeper architectures.
>
> ---
>
> **References**
>
> [1] Vardan Papyan, X. Y. Han, and David L. Donoho. "Prevalence of neural collapse during the terminal phase of deep learning training." Proceedings of the National Academy of Sciences 117.40 (2020): 24652-24663.
>
> [2] Vignesh Kothapalli. "Neural collapse: A review on modelling principles and generalization." Transactions on Machine Learning Research, 2023.
>
> [3] Tomer Galanti, András György, and Marcus Hutter. "Improved generalization bounds for transfer learning via neural collapse." First Workshop on Pre-training: Perspectives, Pitfalls, and Paths Forward at ICML 2022. 2022.
>
> [4] Jingtong Su et al. "On the robustness of neural collapse and the neural collapse of robustness." Transactions on Machine Learning Research, 2024.
>
> [5] Peifeng Gao et al. "Towards demystifying the generalization behaviors when neural collapse emerges." arXiv preprint arXiv:2310.08358 (2023).

---

> > ### Comment · Reviewer_EMH7 · 2025-08-04
> >
> > Thank you for the reply and explanations

---

### Official Review · Reviewer_PA89 · 2025-07-02

**Clarity:** 3
**Significance:** 4
**Originality:** 4
**Rating:** 6
**Confidence:** 3

**Summary:**

This paper addresses a central theoretical question in modern machine learning: Is the prevailing neural collapse globally optimal (in the training data)? The authors prove that NC is globally optimal in deep ResNets and Transformers trained end-to-end with weight decay under cross-entropy or MSE loss. The authors successfully demonstrate that the core mechanism driving collapse is depth and regularization, not architecture details.

The core of their analysis involves a reduction from the end-to-end deep optimization problem to a Generalized Unconstrained Features Model (GUFM). In GUFM, last-layer features are treated as free variables constrained to a unit sphere with zero mean. Previous works proved NC optimality in GUFM for MLPs or in simplified regimes, but this paper rigorously shows that deep ResNets and Transformers approximate GUFM behavior at global optima as depth grows, even when data structure is taken into account.

I particularly appreciate the key conceptual innovation in transformers: by carefully designing the embedding and first attention block, the authors ensure that even with uniform attention weights (zeroed key/query matrices), the necessary context information is encoded into token embeddings. Said another way, Transformers can be shown to approach GUFM-optimal NC solutions using only very weak attention mechanisms.

**Questions:**

1) Is LayerNorm necessary? Could your results be extended to BatchNorm or to ResNets without normalization?
2) Do you anticipate low-rank bias (as discussed in recent work) to appear in your framework under different regularization conditions?

**Ethical Concerns:**

["NO or VERY MINOR ethics concerns only"]

**Final Justification:**

The authors' responses to my questions and other reviewer's questions are convincing. I congratulate the authors for their great work

**Limitations:**

Yes

**Quality:**

4

**Strengths And Weaknesses:**

Strength:

1) Novel theoretical result for ResNets and Transformers:
The first proof of NC optimality in architectures with skip connections, LayerNorm, and attention layers.

2) Rigorous reduction to GUFM:
Bridges the practical training setup of deep networks with unconstrained theoretical models previously considered too simplified.

3) Elegant use of depth as capacity control:
Shows that residual architectures can split large transformations into small norm-preserving steps, enabling global optimization.

4) Unified treatment of CE and MSE loss:
Applies to standard losses used in practice, with theoretical guarantees for both.

Weakness:
1) The assumption of uniform attention might seem unrealistic to readers unfamiliar with theoretical proof constructions. A clarification that this strengthens the claim by showing NC even in the worst-case attention regime would help.

2) Not enough discussion of finite-depth limitations: While convergence rate is discussed (e.g., $O(L^{-1/2})$), readers would benefit from an example

3) For readers not specializing in optimization, could the authors elaborate, in intuitive terms, why the regularization cost in Theorem 4.6 must vanish, whereas in Theorem 4.3 it can remain constant, even though both constructions distribute feature movement across many layers? In particular, can they clarify how the composition of two linear layers per block introduces a “norm floor,” and how this affects the feasibility of reaching GUFM-optimal solutions at finite cost?

---

> ### Author Rebuttal · Authors · 2025-07-30
>
> We thank the reviewer for an insightful review and we are delighted by the reviewer's positive evaluation of our paper. Below, we address questions and concerns.
>
> ---
>
> **Weakness 1. The assumption of uniform attention might seem unrealistic. Please clarify that this strengthens the claim, being the worst-case scenario.**
>
> Thank you for pointing this out. We will explain more clearly in the paper that the uniform attention only serves as a theoretical construction tool and it is not expected to be present in global optima of moderate-depth networks.
>
> ---
>
> **Weakness 2. Provide an example for the convergence rate and discuss the finite depth limitations.**
>
> We have measured the slope of the NC metrics in our experiments of Figures 1 and 2 for architectures with one linear layer per residual connection (computed starting from depth 5, since the shallower depths are too weak to follow the trend), and we have found that its average is at around $-0.335$. This is in agreement with our theory, which predicts a convergence rate between $L^{-1/2}$ and $L^{-1/4}$ (depending on the setting) and, hence, a slope between $-1/2$ and $-1/4$. We will report this observation in the revision.
>
> ---
>
> **Weakness 3. Elaborate, why the regularization cost in Theorem 4.6 must vanish, whereas in Theorem 4.3 it can remain constant.**
>
> This is an important point, which can be best explained through our 1D example briefly discussed in lines 239-244 and 330-333. If we want to fit a constant label, say $\exp(a)$, from an input of size 1, we need to scale our 1D ResNet’s $(1+x)^L$ parameter $x$ as $a/L$, which converges to the label $\exp(a)$ as $L\to\infty$. This linear scaling in $L$ makes the sum of Frobenius norms scale as $1/L.$ However, this is not the case in the second type of ResNet with two linear layers per residual block. This ResNet is of the form $(1+x^2)^L,$ forcing $x$ to scale as  $\sqrt{a}/\sqrt{L}$. This root scaling w.r.t. $L$ makes the sum of squared norms constant w.r.t. $L$. Thus, we need to pick a vanishing regularization term to make the sum of Frobenius norms negligible as the depth grows. The way in which this affects the feasibility of achieving GUFM-optimal solutions at finite cost is explained in Appendix C. There, we give evidence towards the fact that the global optima of the neural network do not approach the global optima of GUFM exactly. To the contrary, the global optima of the neural network are expected to stay within a possibly small, but roughly constant distance from a perfect NC.
>
> We will incorporate the discussion above in the revision.
>
> ---
>
> **Question 1. Is LayerNorm necessary? Do the results hold if BatchNorm was used instead? What about LN-free ResNets?**
>
> This is an interesting question. The LayerNorm is necessary in the first and last layers. It is not necessary in the intermediate layers, though. Therefore, our results hold for ResNets, which only employ the LayerNorm in the embedding and unembedding layers. For ResNets that are fully normalization-free, qualitatively different results have recently been shown in [1], where low-rank bias is demonstrated instead. As for the BatchNorm, we believe our results would still hold, since the presence of normalization is more important than its precise form. However, our proofs would need to be adapted significantly, since they rely on the samples having the same norm throughout the forward pass.
>
> ---
>
>  **Question 2. Do you expect low-rank bias to appear in your setting with different regularization?**
>
> This is a very insightful question. Indeed, as discussed in the previous point, low-rank bias has been demonstrated in some settings for ResNets [1], and it is even responsible for NC not being optimal in MLPs [2]. However, in our setting with LayerNorms and residual connections, a rigorous answer to your question is still open. We do believe, though, that the low-rank bias is *not* present (beyond the low-rankness already guaranteed by neural collapse itself) in this setting. The intuition is that low-rank bias is mostly present in deep architectures because maintaining high-rank features requires high-rank weight matrices, which is costly in terms of Frobenius norm. However, in the ResNets and transformers we consider here, maintaining high-rank features can be done for free by setting the weight matrices to 0. From this point of view, there is no obvious reason for low-rank bias to hold in our setting.
>
> ---
>
> **References**
>
> [1] Enric Boix-Adsera. "On the inductive bias of infinite-depth ResNets and the bottleneck rank." arXiv preprint arXiv:2501.19149 (2025).
>
> [2] Peter Súkeník, Christoph Lampert, and Marco Mondelli. "Neural collapse vs. low-rank bias: Is deep neural collapse really optimal?." Advances in Neural Information Processing Systems 37 (2024): 138250-138288.

---

> > ### Comment · Reviewer_PA89 · 2025-08-06
> > **Thank you for your response**
> >
> > I read through the authors' rebuttal to my questions and other reviewer's questions. i am generally satisfied and I will maintain my recommendation for acceptance

---

### Official Review · Reviewer_s3Hv · 2025-07-03

**Clarity:** 2
**Significance:** 3
**Originality:** 3
**Rating:** 5
**Confidence:** 3

**Summary:**

The paper establishes a theoretical connection between the generalized unconstrained feature model (GUFM) and deep post-LayerNorm residual networks and Transformers.

The authors first show that all global optima of GUFM, under CE and MSE losses, exhibit vanishing neural collapse metrics. They then prove that sufficiently deep L-RN1 and L-Tx1 networks can approximate any such GUFM solution, and hence the collapse properties. Later on, the authors extend their analysis to show in L-RN2 and L-Tx2 (2 layers in each MLP block) also share the same property as the regularization strength goes to 0.

Empirical results incorporated in the paper demonstrate that as depth grows, neural collapse metrics consistently decrease.

**Questions:**

* Is it possible to design some experiments to show that the constructed optima in the paper are achievable?
* It seems like the NC metrics shown in the experiment section are not very small given the very long training time (5000) epoch. What would be the cause of this? Based on my previous experience, it would be better to include some kind of learning rate annealing techniques to encourage NC. I wonder whether the observed results still hold when adding this kind of more subtle optimization strategy.

**Ethical Concerns:**

["NO or VERY MINOR ethics concerns only"]

**Final Justification:**

This is an interesting paper within the Neural Collapse community, and I recommend acceptance. Given the complexity of the problem being analyzed, I would think the use of constructed solutions (while questioned by several reviewers, including myself) to be a reasonable choice.

**Limitations:**

Yes

**Quality:**

3

**Strengths And Weaknesses:**

Strengths:
* The focus of the paper covers a very interesting yet not well-established theoretical question. Many previous NC theoretical results focuses on proving using UFM but the connection between UFM and modern deep networks remain elusive. This paper steps towards closing this gap.
* The thretical results are quite comprehensive, cover both residual based network and Transformers, also consists of different architecture choice within these models.

Weaknesses:
* The notation is a bit un-organized, and it hinders the readibility of the paper. For example, $lin$ is defined as a affine transformation but in the last layer $lin_L$, the bias term is dropped.
* The global optima constructed in the paper is not verified: we don't know whether gradient based optimization method would converge to such solutions, it would be nice to design some experiments to check this.
* Related to the previous point, the paper does not address whether all global minima exhibit neural collapse, this would something valuable to check perhaps in future work.

---

> ### Author Rebuttal · Authors · 2025-07-30
>
> We thank the reviewer for the comments and the overall positive evaluation. We address concerns and questions below.
>
> ---
>
> **Weakness 1. Unorganized notation.**
>
> Thanks for pointing out the notation inconsistency. We will make this explicit in the revision.
>
> ---
>
> **Weaknesses 2. Are global optima constructed in the paper achievable by gradient descent?**
>
> We note that the optimality we show is in the limit $L\to\infty$: our result proves neural collapse for global optima as $L\to\infty$. This means that, when $L$ grows, as long as gradient descent finds a global optimum, the penultimate layer of the gradient descent solution will necessarily coincide with that of the solution we propose. Now, there may be different ways to converge to the limit. In the paper we describe one that is convenient to analyze, in which samples are moved one at a time, i.e., different layers are used to move different samples. Gradient descent may choose a different solution and, in fact, we would expect architectures trained via gradient-based optimization to move several samples simultaneously.
>
> ---
>
> **Weakness 3. Do all global optima exhibit neural collapse?**
>
> Yes, Corollaries 4.5 and 4.7 show that **all** global optima exhibit approximate neural collapse (Corollary 4.5 is for deep single-layer architectures, and Corollary 4.7 is for deep double-layer architectures). Thus, **any** solution which does not exhibit neural collapse cannot be a global optimum.
>
> ---
>
> **Question 1. Is it possible to design some experiments to show that the constructed optima in the paper are achievable?**
>
> As mentioned above, we do not necessarily expect the constructed solutions to be achievable by gradient descent. While our construction moves one sample at a time, we expect solutions obtained via gradient descent to be more efficient in the way the network uses the depth. However, we note that the convergence rate predicted by our theory is in line with what we observe in numerical experiments. In fact, the slopes of the NC curves in Figures 1 and 2 for networks with one linear layer per residual block (computed starting from depth 5, since the shallower depths are too weak to follow the trend) average at around -0.335. This is comparable to our theory, which ranges between $-1/2$ and $-1/4$, depending on the setting.
>
> We also note that the main takeaway of our proof remains true in practice: as the depth grows, layers get smaller, splitting the total work, so that their total norm is decreased. To support this claim, we have computed the sum of all layers’ squared Frobenius norms multiplied by a constant regularization parameter for ResNet trained on MNIST (corresponding to the experiment of Figure 2 in the Appendix), as a function of the depth of the network.
>
> Number of layers: [2, 3, 5, 8, 13, 21, 34]
>
> Means: [0.1467, 0.1471, 0.1360, 0.1182, 0.0986, 0.0835, 0.0711]
>
> Standard deviations: [0.0015, 0.0014, 0.0004, 0.0005, 0.0010, 0.0006, 0.0002]
>
> This clearly shows that, as the number of layers increases, the total norm of the weights decreases. We note that this is the same phenomenon leading to neural collapse in our theory. Similar results hold for the other training scenarios as well. We will add a figure in the revision.
>
> ---
>
> **Question 2. The NC metrics are not very small, considering the large training time. Why? Would LR annealing help?**
>
> The main reason why the NC metrics are not very small for shallower networks is that the collapse is not *exactly* optimal in shallow ResNets and transformers. However, as our theory predicts, the precise collapse gets approached more and more closely by global optima as we increase the depth. It is visible from Figure 1 that this is indeed the case. Moreover, the slopes of the NC curves in our experiments are comparable to our theory, see the answer to the previous question. We believe LR annealing techniques would not help decrease NC metrics further, because we already employ a ten-fold LR drop at epoch 4000 in all our experiments, and the NC formation is not heavily affected by this (in fact, it is slowed down, as opposed to being speeded up). We will clarify this point in the revision.

---

> ### Comment · Reviewer_s3Hv · 2025-08-05
>
> I thank the authors for their response and clarifications. I remain positive about the paper. Given the complexity of the problem being analyzed, I would think the use of constructed solutions (while questioned by several reviewers, including myself) to be a reasonable choice.

---

### Decision · Program_Chairs · 2025-09-17

**Decision:**

Accept (poster)

**Comment:**

This paper investigates neural collapse (NC) in deep regularized ResNets and transformers with LayerNorm, trained end-to-end using cross-entropy or mean squared error loss. The authors prove that the global optima of these architectures exhibit approximate NC as depth increases.

The theoretical analysis extends prior work on the unconstrained features model (UFM) and MLP-focused studies. Reviewers highlighted that the rigorous reduction of end-to-end training to a generalized UFM (GUFM) provides an elegant bridge from complex architectures to simpler models, yielding novel insights into NC optimality. On the other hand, the analysis relies on constructive proofs, and therefore provides limited explanation of why NC emerges under gradient descent dynamics in practice. Overall, the reviewers consensually recognize the paper as a technically solid and insightful theoretical contribution.